# Nestin regulates cellular redox homeostasis in lung cancer through the Keap1–Nrf2 feedback loop

Jiancheng Wang[1,2,11], Qiying Lu[1,2,11], Jianye Cai[2,3,11], Yi Wang[1,2], Xiaofan Lai[4], Yuan Qiu[1,2], Yinong Huang[2,5], Qiong Ke[1,2], Yanan Zhang[1,2], Yuanjun Guan[6], Haoxiang Wu[2], Yuanyuan Wang [2], Xin Liu[2], Yue Shi[2], Kang Zhang[7], Maosheng Wang[8]* & Andy Peng Xiang[1,2,6,9,10]*

Abnormal cancer antioxidant capacity is considered as a potential mechanism of tumor malignancy. Modulation of oxidative stress status is emerging as an anti-cancer treatment. Our previous studies have found that Nestin-knockdown cells were more sensitive to oxidative stress in non-small cell lung cancer (NSCLC). However, the molecular mechanism by which Nestin protects cells from oxidative damage remains unclear. Here, we identify a feedback loop between Nestin and Nrf2 maintaining the redox homeostasis. Mechanistically, the ESGE motif of Nestin interacts with the Kelch domain of Keap1 and competes with Nrf2 for Keap1 binding, leading to Nrf2 escaping from Keap1-mediated degradation, subsequently promoting antioxidant enzyme generation. Interestingly, we also map that the antioxidant response elements (AREs) in the Nestin promoter are responsible for its induction via Nrf2. Taken together, our results indicate that the Nestin–Keap1–Nrf2 axis regulates cellular redox homeostasis and confers oxidative stress resistance in NSCLC.

[1] Program of Stem Cells and Regenerative Medicine, Affiliated Guangzhou Women and Children's Hospital, Zhongshan School of Medicine, Sun Yat-Sen University, Guangzhou, China. [2] Center for Stem Cell Biology and Tissue Engineering, Key Laboratory for Stem Cells and Tissue Engineering, Ministry of Education, Sun Yat-Sen University, Guangzhou, China. [3] Department of Hepatic Surgery and Liver Transplantation Center of the Third Affiliated Hospital, Organ Transplantation Institute, Sun Yat-Sen University, Guangzhou, China. [4] Department of Anesthesiology, The First Affiliated Hospital of Sun Yat-sen University, Guangzhou, China. [5] Department of Neurology, The Third Affiliated Hospital of Sun Yat-Sen University, Guangzhou, China. [6] Core Facility of Center, Zhongshan School of Medicine, Sun Yat-Sen University, Guangzhou, China. [7] Faculty of Medicine, Macau University of Science and Technology, Macau, China. [8] The Cardiovascular Center, Gaozhou People's Hospital, Maoming, China. [9] Department of Biochemistry, Zhongshan School of Medicine, Sun Yat-Sen University, Guangzhou, China. [10] Key Laboratory of Protein Modification and Degradation, School of Basic Medical Sciences, Affiliated Cancer Hospital and Institute of Guangzhou Medical University, Guangzhou, China. [11]These authors contributed equally: Jiancheng Wang, Qiying Lu, Jianye Cai. *email: mmwmsmd@126.com; xiangp@mail.sysu.edu.cn

I t is well known that an imbalance between reactive oxygen species (ROS) generation and elimination contributes to the moderate oxidative stress commonly seen in cancer[1]. Cancer cells characteristically have higher ROS levels than normal cells due to mitochondrial dysfunction or metabolic abnormality[2,3], and thus develop powerful antioxidant defenses that modulate ROS to levels that are suitable for cancer initiation and transformation[4]. Therefore, targeting the antioxidant capacity of cancer cells might have a beneficial therapeutic impact.

Numerous regulators are known to have significant impacts on intracellular antioxidant defenses. The transcription factor, NF-E2-related factor 2 (Nrf2), is considered to be a master regulator of the expression levels of various antioxidant enzymes, including glutathione S-transferase (GST), NAD(P)H:quinone oxidoreductase 1 (NQO1), and others, via binding enhancer sequences termed "antioxidant-response elements" (AREs)[5,6]. In addition to its ability to improve the antioxidant capacity, activated Nrf2 remodels the metabolic reprogramming by redirecting glucose and glutamine to the anabolic pathway, and also influences apoptosis by associating with p62/SQSTM1[7,8]. It has been demonstrated that the constitutive stabilization and activation of Nrf2 is associated with poor prognosis in various human cancers, such as hepatocellular carcinomas, lung cancer, and gallbladder cancer[9]. However, the mechanisms by which Nrf2 promotes malignancy have not been fully explored.

Under resting physiological conditions, Nrf2 activity is tightly restricted by its binding with the Kelch-like ECH-associated protein 1 (Keap1)-Cullin 3 (Cul3) E3-Rbx1 ligase complex in the cytoplasm, which limits its translocation from the cytosol to the nucleus. As a consequence, very low constitutive levels of Nrf2 are responsible for maintaining the basal antioxidant levels[6]. Mechanistic studies have revealed that Keap1 functions as a key scaffold in Cul3-containing E3 ubiquitin ligase[10]. Under oxidative stress, chemopreventive compounds ($H_2O_2$, $O_2^-$, and so on) inhibit the activity of Keap1-Cul3-Rbx1 E3 ubiquitin ligase, contributing to the increased levels of Nrf2 and the activation of its downstream target genes. Once cellular redox homeostasis has been recovered, Keap1 translocates into the nucleus and releases Nrf2 from the AREs. This redox stress-sensing adaptive response system has been studied widely in terms of its molecular mechanisms and biological significance[11]. On one hand, modification of a cysteinyl residue (151AA) of Keap1 renders it unable to ubiquitinate Nrf2[12]. On the other hand, disruption of Keap1–Nrf2 complex stability could also affect Nrf2 protein levels[13]. However, we do not yet know the details of how this pathway responds to oxidative stress. In particular, it would be interesting to elucidate the factor(s) responsible for regulating the binding between Nrf2 and Keap1. Several factors, such as iASPP[14] and p62/SQSTM1[7,8], DPP3[15,16], PALB2[17,18] have been identified as the members of Nrf2–Keap1 stress signaling hub, but neither is exclusive. Thus, the other proteins engaged in the Keap1 pathways and the potential mechanism underlying the dynamic of Nrf2–Keap1 still remain to be further explored.

A growing body of evidence demonstrates that intermediate filament proteins, which form a class of important cellular stress proteins, help protect cells from a variety of stresses and contribute to maintaining intracellular redox homeostasis[19]. Nestin is a class VI intermediate filament protein that is extensively expressed in tumors and stem cells. Sahlgren et al. found that downregulation of Nestin sensitized neuronal progenitor cells to exogenous ROS-induced cell death[20], suggesting that Nestin may serve as a survival determinant during oxidative stress. Moreover, our group found that Nestin colocalized with mitochondria, altered mitochondrial dynamics and functions by assisting with the mitochondrial recruitment of Dynamin-related protein1 (Drp1), and thereby influences the intracellular redox status[21]. In the same study, we also observed that the antioxidant capacity of cancer cells decreased significantly upon ablation of Nestin expression, indicating that Nestin might participate in the regulation of oxidative stress. Whether Nestin is an antioxidative factor or is involved in the Nrf2–Keap1–ARE signaling pathway needs to be elucidated.

In the present study, we reveal that Nestin competitively combines with the Kelch domain of Keap1 to protect Nrf2 against Keap1-mediated degradation, and subsequently upregulates the expression of antioxidant enzymes. Interestingly, Nrf2 directly binds to the ARE motifs of the Nestin promoter and induces Nestin expression under oxidative stress to form a positive-feedback loop. Taken together, our findings suggest that targeting the Nestin–Keap1–Nrf2 signaling pathway could be a promising therapeutic approach for inhibiting malignant initiation and progression.

## Results

**Nestin knockdown reduces antioxidant capacity of NSCLC cells.** To examine the relationship between Nestin and the antioxidant capacity of cells, we used Nestin-short hairpin RNAs (shNestin1 or shNestin2) to specifically reduce Nestin expression in NSCLC cell lines (Fig. 1a and Supplementary Fig. 1a). We used Annexin V/propidium iodide (PI) flow cytometric analysis to measure the effect of Nestin knockdown on cell death among A549 and H1299 cells. Nestin knockdown alone had little effect on the rate of apoptosis (Supplementary Fig. 1b, c), but $H_2O_2$-induced cell apoptosis was significantly increased by Nestin knockdown (Fig. 1b, c). A LIVE/DEAD viability/cytotoxicity assay confirmed that Nestin-knockdown NSCLC cells were more sensitive to $H_2O_2$-induced cell death (Supplementary Fig. 1d, e). These results suggested that Nestin knockdown decreased the antioxidant capacity of NSCLC cells. To explore whether Nestin influenced antioxidant capacity of these cells, we examined the gene expression levels of various antioxidant proteins, such as catalase (CAT), glutathione peroxidase 1 (GPX1), glutathione peroxidase 4 (GPX4), superoxide dismutase 1 (SOD1), superoxide dismutase 2 (SOD2), glutamate-cysteine ligase, catalytic subunit (GCLC), glutamate-cysteine ligase modifier subunit (GCLM), heme oxygenase 1 (HMOX-1), and NQO1. We found that Nestin knockdown decreased the mRNA levels of all these genes in NSCLC cells (Fig. 1d, e). In addition, the levels of the antioxidant, glutathione (GSH), the activities of SOD and CAT, and the total antioxidant capacity were all reduced upon Nestin knockdown (Fig. 1f–h and Supplementary Fig. 1f). In addition, Nestin overexpression by transfection with Myc-Nestin vectors significantly enhanced the expression of antioxidant genes (Supplementary Fig. 1g, h), as well as the levels of GSH, the activities of SOD and CAT (Supplementary Fig. 1i–k).

To further determine whether Nestin could regulate the antioxidant capacity in vivo, we used an inducible RNA interference (RNAi) xenograft model. An inducible RNAi system comprising regulatory and response plasmids was introduced into NSCLC cells via lentiviral transfection (Fig. 1i and Supplementary Fig. 2a), and the transfected NSCLC cells were subcutaneously injected into nude mice. For induction of Nestin knockdown, mice were treated with doxycycline (Dox) via their food pellets after 2 weeks. The results showed that tumor growth rates and volumes were significantly reduced upon Dox administration (Supplementary Fig. 2b). Then, we used qPCR and IHC staining to confirm that Dox treatment successfully reduced Nestin expression (Fig. 1j, k). Subsequent experiments revealed that the Dox-treated group exhibited decreases of various antioxidant molecules (Fig. 1l and Supplementary Fig. 2c–e). Furthermore, ROS staining revealed that the xenograft tumors had higher ROS

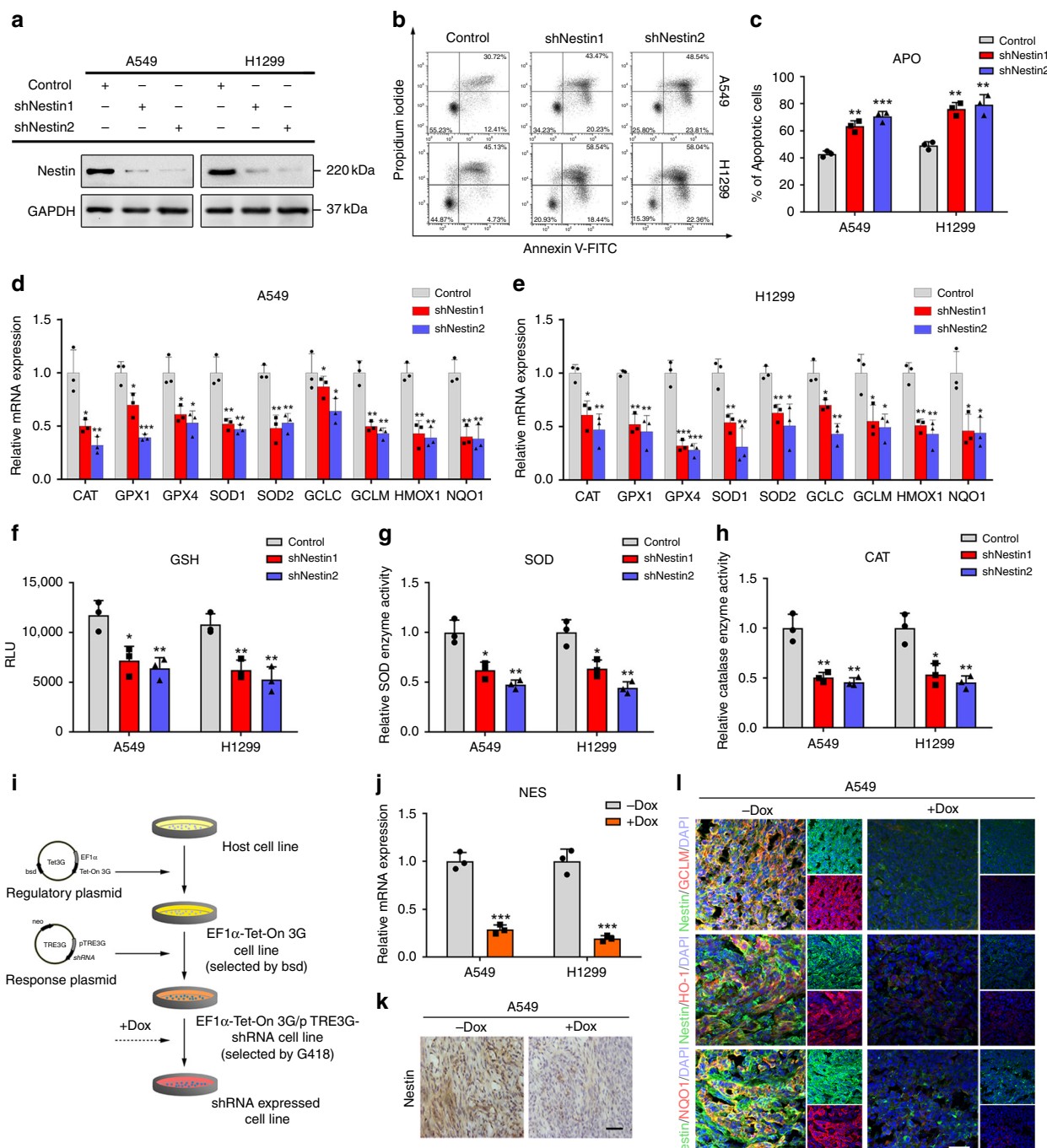

**Fig. 1** Nestin knockdown reduces the antioxidant capacity in NSCLC cells. **a** Nestin was downregulated using specific shRNAs (shNestin1 and shNestin2). At 72 h post-transfection, whole-cell extracts were prepared and Nestin levels were analyzed by Western blotting. **b** Flow cytometric detection of $H_2O_2$-induced NSCLC cell death was achieved using Annexin V-FITC and propidium iodide (PI). **c** Statistical analysis of the total apoptosis rate in NSCLC cells. **d**, **e** qPCR analysis showing that knockdown of Nestin reduced the expression levels of several antioxidation-related genes compared to those seen in control cells. **f** Analysis of GSH levels in NSCLC cells. **g** SOD activity in NSCLC cells was examined with a SOD assay kit. **h** Analysis of CAT levels in NSCLC cells. **i** The Tet-On 3G doxycycline-inducible gene expression system was used to control the translation of a human Nestin-targeting shRNA driven by the TRE3G promoter (PTRE3G) in NSCLC cells. Briefly, NSCLC cells were transfected with the EF1α-Tet-On 3G-bsd plasmid and selected with blasticidin. The cells were then transfected with the pTRE3G-shRNA-neo plasmid, in which the shRNA was incorporated downstream of Tet-regulated PTRE3G. The cells were selected with neomycin, and shRNA expression was induced in surviving cells with doxycycline (Dox) treatment. **j** The ability of the Tet-On 3G doxycycline-inducible gene expression system to downregulate Nestin in NSCLC cells in vitro was confirmed by qPCR. **k** IHC staining indicating that Nestin expression was downregulated in xenograft tumors following doxycycline treatment in vivo. Briefly, A549 cells transfected with EF1α-Tet-On 3G-bsd and pTRE3G-shRNA-neo plasmids were subcutaneously injected into nude mice. Two weeks later, mice were fed diets with or without Dox (625 mg/kg food). Xenograft tumors were collected 5 weeks after grafting, and IHC staining was performed. Scale bar: 50 μm. **l** Representative images of immunofluorescence. Xenograft tumor samples were obtained as described in **k** and labeled with anti-GCLM/HO-1/NQO1 (red), anti-Nestin (green) and 4′,6-diamidino-2-phenylindole (DAPI; blue). Scale bar: 50 μm. Data are presented as the means ± SD of three independent experiments. *$P < 0.05$, **$P < 0.01$ and ***$P < 0.001$, Student's $t$-test. Source data are available as a Source Data file

levels in the Dox-treated group (Supplementary Fig. 2f). These results indicated that the Dox-induced knockdown of Nestin could decrease the antioxidant capacity of NSCLC cells in vivo. Taken together, our results suggest that Nestin plays an important role in regulating the intracellular antioxidant system.

**Nestin knockdown inactivates the Nrf2–ARE pathway.** As Nrf2 is widely accepted to be a key transcription factor responsible for regulating the antioxidant defense systems, we questioned whether Nestin influences antioxidant protein expression through regulating the Nrf2 signaling pathways. Indeed, we found that Nestin knockdown decreased Nrf2 expression in NSCLC cells (Fig. 2a, b). As CAT, GPX1, GPX4, SOD1, SOD2, GCLC, GCLM, HMOX-1 and NQO1 are transcriptional targets of Nrf2, which binds the AREs in their promotor regions[6], we hypothesized that Nestin might regulate the expression of these antioxidant molecules via the Nrf2–ARE signaling pathway. To test this possibility, we transfected NSCLC cells with an ARE luciferase reporter. Indeed, we found that Nestin knockdown significantly suppressed the activity of the luciferase reporter (Fig. 2c). Moreover, immunoblotting revealed that the Nestin knockdown decreased the protein levels of NQO1, GCLM, and HO-1 (Fig. 2d).

To assess the ability of Nestin to associate with Nrf2, we treated cells with the Nrf2 activators, tert-butylhydroquinone (tBHQ) and sulforaphane (SF)[22]. tBHQ and SF enhanced the protein levels of Nrf2 in both cell lines, but the levels of Nrf2 were lower in Nestin-knockdown cells compared to control cells (Fig. 2e and Supplementary Fig. 3a). Moreover, the transcriptional levels of Nrf2-downstream genes were reduced in tBHQ-treated or SF-treated Nestin-knockdown cells (Fig. 2f–h and Supplementary Fig. 3b–d). These results suggest that Nestin may regulate Nrf2 expression under both basal and induced conditions. To further verify that the antioxidant function of Nestin was mediated through the upregulation of Nrf2 signaling, we performed ARE luciferase reporter assays and apoptosis assays. In Nrf2-knockdown cells, Nestin knockdown had little effect on ARE luciferase expression (Fig. 2i), the expression levels of HMOX1, GCLM, and NQO1 (Supplementary Fig. 3e, f), or oxidative stress-induced cell death (Fig. 2j, k). These results indicate that Nestin regulates the antioxidant system by stabilizing Nrf2 protein levels and subsequently upregulating Nrf2–ARE signaling.

**Nestin prevents the degradation of Nrf2 protein.** As intracellular protein levels are determined by the balance between protein synthesis and degradation, we examined these parameters of Nrf2 in NSCLC cells with or without Nestin knockdown. qPCR analysis revealed that Nestin knockdown had no effect on the transcription of Nrf2 in NSCLC cells (Fig. 3a), prompting us to speculate that Nestin knockdown might impact the degradation of Nrf2. Accordingly, we treated NSCLC cells with the protein synthesis inhibitor, cycloheximide (CHX), and analyzed the degradation rate of Nrf2. Indeed, we found that the degradation of Nrf2 was accelerated upon Nestin knockdown in NSCLC cells under both basal and induced conditions, respectively (Fig. 3b, c and Supplementary Fig. 4a, b). To examine whether Nestin knockdown accelerated Nrf2 degradation by increasing its ubiquitination via the ubiquitin–proteasome pathway, we treated NSCLC cells with the proteasome inhibitor, MG132. We found that MG132 treatment rescued Nrf2 protein levels in Nestin-knockdown cells (Fig. 3d and Supplementary Fig. 4c). Moreover, a ubiquitination assay showed that the basal levels of ubiquitin-conjugated Nrf2 were increased in Nestin-knockdown cells, whereas Nestin overexpression reduced the levels of ubiquitin-conjugated Nrf2 (Fig. 3e). Nrf2 is activated by its dissociation from an inactive complex in the cytoplasm and subsequent

translocation into the nucleus. Interestingly, we found that Nestin-knockdown cells had higher levels of cytoplasmic Nrf2, and that its nuclear translocation was rescued by the re-expression of Nestin (Fig. 3f–g and Supplementary Fig. 4d). According to the observation that Nestin regulated the distribution of Nrf2, we then asked whether Nestin might directly bind to Nrf2 and decrease its degradation. However, the co-immunoprecipitation assays showed that Nestin did not directly interact with Nrf2 in NSCLC cells (Fig. 3h). Collectively, these findings suggest that Nestin can stabilize Nrf2 by indirectly preventing its ubiquitin–proteasome-mediated degradation.

**Nestin competes with Nrf2 for Keap1 binding.** Keap1 is well known to act as a substrate adaptor to bring Nrf2 into the Cul3-dependent E3 ubiquitin ligase complex, resulting in the rapid proteasome-mediated degradation of Nrf2[23,24]. We thus explored the effect of Nestin knockdown on the expression of the Keap1–Cul3 complex. We found that Nestin knockdown had no effect on Keap1 expression at the mRNA and protein levels (Fig. 4a, b), nor did it alter the ubiquitination of Keap1 or the protein levels of Cul3 (Fig. 4c and Supplementary Fig. 4e). Therefore, we investigated whether Nestin prevented the degradation of Nrf2 by interacting with Keap1. Our immunoprecipitation assay clearly showed that Nestin directly bound to Keap1 (Fig. 4d, e). Using super-resolved fluorescence microscopy, we further confirmed that Keap1 and Nestin colocalized throughout the cells (Fig. 4f). We also performed an immunoprecipitation assay using MG132-treated A549 cells and found that Keap1 bound more ubiquitined-Nrf2 after Nestin knockdown (Fig. 4g). The above results suggest that Nestin competitively binds to Keap1, thereby inhibiting the Keap1–Nrf2 interaction and subsequent Nrf2 degradation.

**The ESGE motif in Nestin binds the Kelch domain of Keap1.** To test how Nestin competitively bound to Keap1, we constructed a series of Nestin and Keap1 deletion mutants and co-expressed a series of truncated Nestin proteins in HEK293FT cells along with Flag-tagged Keap1 (Fig. 5a). Immunoprecipitation assays showed that Flag-Keap1 specifically interacted with the full-length (N1-1621) and C-terminal tail domain-containing fragments (N641-1621 and N1295-1621) of Myc-Nestin, indicating that N1295-1621 of Nestin might mediate the interaction with the Keap1 protein (Fig. 5b). To map which domain of Keap1 was required for Nestin binding, reciprocal immunoprecipitation assays revealed that only the Kelch domain-containing fragments bound to Myc-Nestin (Fig. 5c), suggesting that Keap1 associates with Nestin through the Kelch domain (N322-609) of Keap1.

To further detect whether Nestin protein fragment N1295-1621 was sufficient to stabilize Nrf2 protein levels and enhance Nrf2–ARE signaling, we used Myc-Nestin as positive control. Indeed, an ARE luciferase reporter assay showed that ARE activity in Nestin-knockdown cells was rescued by transfection with Nestin protein fragment N1295-1621 (Fig. 5d). Consistent with this finding, Nestin fragment N1295-1621 could also rescue the expression of Nrf2 downstream genes in Nestin-knockdown cells (Fig. 5e, f and Supplementary Fig. 5a). Furthermore, Nestin fragment N1295-1621 could reverse the decreased antioxidant capacity in Nestin-knockdown cells, as assessed by increases in SOD activity, the GSH level and the total antioxidant capacity (Fig. 5g, Supplementary Fig. 5b, c), and this was associated with a significant decrease in $H_2O_2$-induced cell death (Fig. 5h, i). These data collectively demonstrate that Nestin directly binds with Keap1, and that this association might contribute to the cellular responses to oxidative stress.

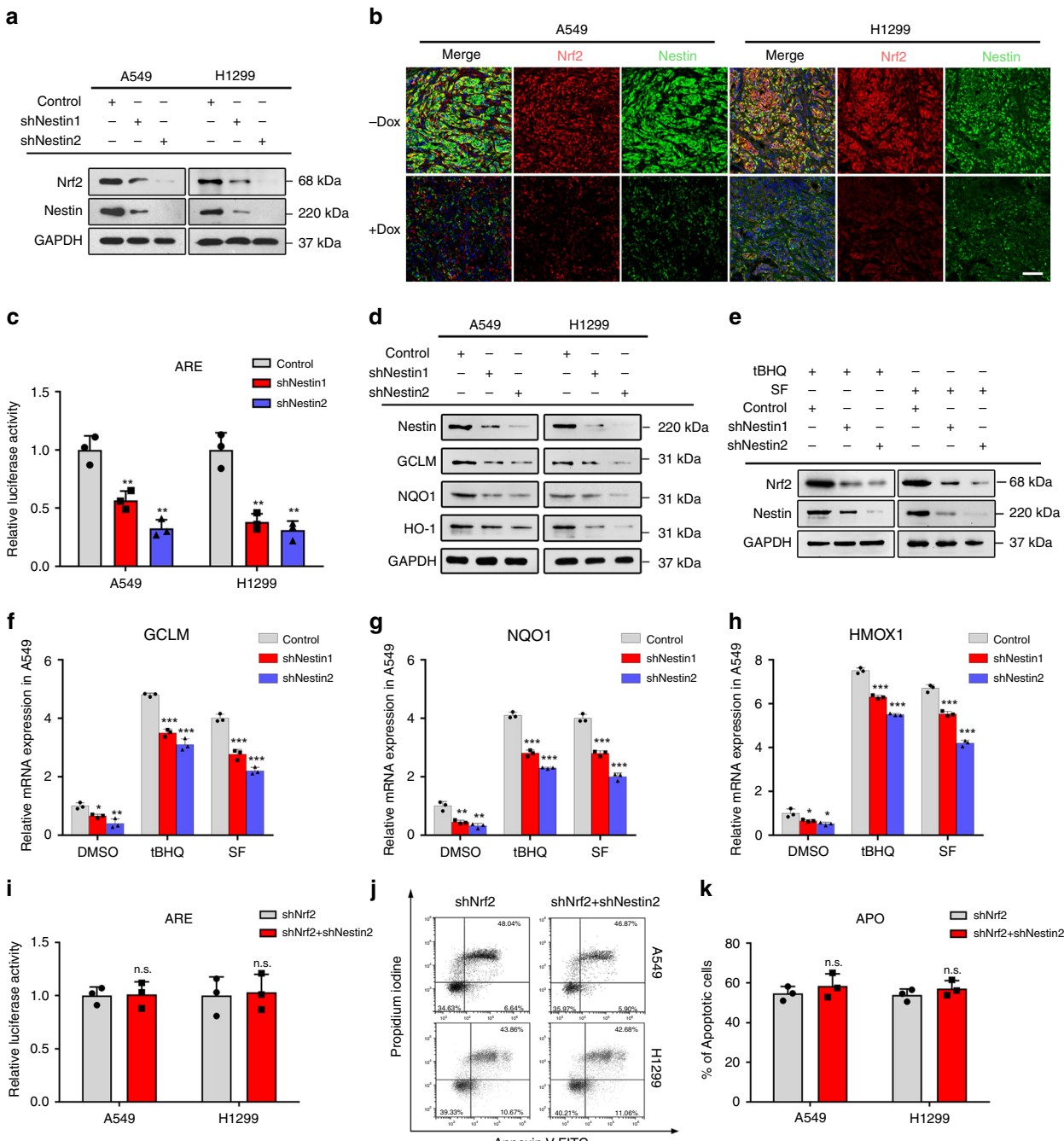

**Fig. 2** The impact of Nestin on cellular antioxidant activity is mediated by the Nrf2–ARE pathway. **a** Immunoblotting was performed to detect the expression levels of Nestin and Nrf2 in NSCLC cells with or without Nestin knockdown. **b** Immunostaining was performed on sections of the xenograft tumors using the Tet-On 3G doxycycline-inducible gene expression system. The xenograft tumor samples were labeled with anti-Nrf2 (red), anti-Nestin (green), and DAPI (blue). Scale bar: 50 μm. **c** A luciferase assay was used to detect reporter gene activity from the AREs. NSCLC cells with or without Nestin knockdown were transiently transfected with an ARE luciferase reporter plasmid. At 24 h post-transfection, the cells were assayed for luciferase activity. The results are expressed as the fold-change of luciferase activity with respect to that of the vector control. **d** Western blot analysis representing the protein levels of NQO1, HO-1, and GCLM in NSCLC cells with or without Nestin knockdown. **e** The basal and induced protein levels of Nrf2 were lower in A549 cells with Nestin knockdown. NSCLC cells were treated with 100 μM tBHQ or 20 μM SF for 16 h to activate Nrf2. **f–h** The mRNA levels of Nrf2-downstream genes were analyzed by qPCR in Nestin-knockdown A549 cells. A549 cells were treated with DMSO (as a control), 100 μM tBHQ, or 20 μM SF for 16 h. **i** Nrf2-knockdown NSCLC cells transfected with vector or shNestin plasmids were transfected with the ARE luciferase reporter. At 24 h post-transfection, the cells were assayed for luciferase activity. **j** Nrf2-knockdown A549 cells transfected with vector or shNestin plasmids were treated with 200 μM $H_2O_2$ for 6 h, and flow cytometric detection of apoptosis was performed using Annexin V-FITC and PI. **k** Statistical analysis of the total apoptosis rates of the A549 cells described in **j**. The total apoptosis rates were calculated as the sum of the early and late apoptosis rates. Data are presented as the means ± SD of three independent experiments. **P < 0.01, N.S. represents no significant, Student's t-test. Source data are available as a Source Data file

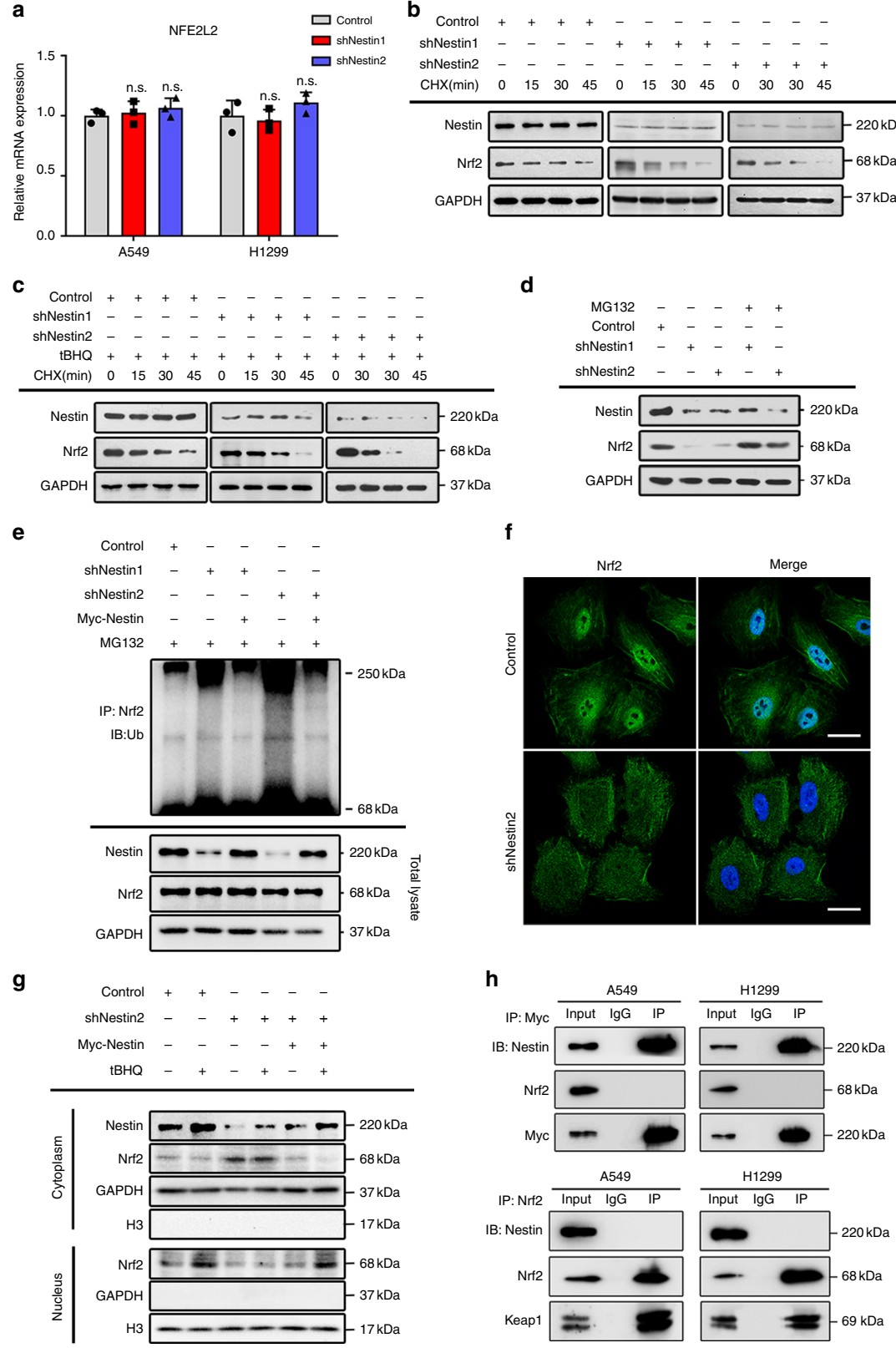

Previous studies demonstrated that Keap1-associated proteins (e.g., Nrf2 and PGAM5) contain a consensus E(S/T)GE motif that is for binding to the Kelch domains of Keap1[13]. Interestingly, the Nestin proteins of different species share a highly conserved ESGE motif in their C-terminal sequences (Fig. 5j). Then, we constructed an ESGE-deletion vector (ΔESGE) and a missense mutant vector

(ESGA) in which ESGE[1417] was changed to ESGA[1417]. As shown in Fig. 5k, both of these Nestin mutants were unable to associate with Keap1. Immunoprecipitation showed that the basal levels of ubiquitin-conjugated Nrf2 were increased in ΔESGE and ESGA cells (Fig. 5l), suggesting that the ESGE motif of Nestin is essential for its competitive binding of Keap1. Furthermore, overexpression

**Fig. 3** Nestin protects Nrf2 from ubiquitin–proteasome degradation. **a** The mRNA levels of the Nrf2-encoding gene (*NFE2L2*) did not significantly differ in NSCLC cells transfected with Nestin-knockdown and control plasmids. **b** Nestin stabilized Nrf2 under basal conditions. A549 cells transfected with Nestin-knockdown and control plasmids were left untreated or treated with 50 μg/mL CHX and incubated for the indicated time periods. Lysates were analyzed by Western blotting. **c** Nestin stabilized Nrf2 under stress conditions. A549 cells with or without Nestin knockdown were pretreated with 100 μM tBHQ for 4 h, treated with 50 μg/mL CHX and incubated for the indicated durations. **d** Nestin reduced the protein degradation of Nrf2. A549 cells transfected with Nestin-knockdown and control plasmids were left untreated or treated with 10 μM of MG132 for 4 h to block the degradation of ubiquitinated proteins. **e** Nestin reduced the ubiquitination of Nrf2. Control or Nestin-knockdown NSCLC cells were transfected with or without Myc-Nestin plasmids, treated with 10 μM of MG132 for 4 h and subjected to an in vivo ubiquitination assay to detect Ubiquitin-conjugated endogenous Nrf2 proteins. Lysates were denatured, immunoprecipitated with anti-Nrf2 and blotted with an anti-Ubiquitin antibody. **f** Immunofluorescence was used to localize Nrf2 in cells with or without Nestin knockdown. NSCLC cells were labeled with anti-Nrf2 (green) and DAPI (blue). Scale bar: 20 μm. **g** Knockdown of Nestin increased the nuclear translocation of Nrf2, and this effect could be rescued by overexpression of Nestin. NSCLC cells transfected with control, Nestin-knockdown (shNestin2), or Nestin-overexpression (Myc-Nestin) plasmids were treated with or without 100 μM tBHQ for 4 h. Subcellular fractionation was used to isolate cytoplasmic and nuclear proteins, and immunoblotting was performed to examine the localization of Nrf2 following the downregulation or overexpression of Nestin. **h** Nestin did not directly interact with Nrf2. Myc-Nestin was transfected into NSCLC cells. Whole-cell lysates were immunoprecipitated with anti-Myc antibodies and the precipitated proteins were blotted with the indicated antibodies. N.S. represents no significant, Student's *t* test. Source data are available as a Source Data file

of Nestin ΔESGE in Nestin-knockdown cells was incapable of rescuing ARE activity, Nrf2 downstream gene expression and the antioxidant capacity (Supplementary Fig. 5d–i). Taken together, these results demonstrate that the ESGE motif of Nestin is responsible for its competitive binding to Keap1, which subsequently protects Nrf2 from degradation.

**Nrf2 promotes the transcription of Nestin**. Our above data identify that Nestin can help maintain the redox balance by regulating the antioxidant capacity in NSCLC cells. We previously found that oxidative stress could enhance Nestin expression in NSCLC cells in vitro. However, we did not know how the intracellular oxidative status influenced Nestin expression. To explore the factors that could mediate the transcription of Nestin under conditions of oxidative stress, we analyzed the human Nestin gene promoter for transcription factor-binding sequences related to the oxidative stress response and found several conserved AREs (5′-RTG AYnnnGCR-3′) located 10 kb upstream from the transcriptional start site of Nestin (Supplementary Table 1). Further analysis identified that these ARE sequences were conserved in the Nestin promoters of various species (Fig. 6a). To test whether Nrf2 regulated Nestin transcription by binding to the AREs located in the Nestin promoter, NSCLC cells were treated with the Nrf2 activator, tBHQ, we found that the luciferase activity were enhanced by hARE1 and hARE2, rather than hARE3 of the Nestin promoter (Fig. 6b). Furthermore, we performed electrophoresis mobility shift assays (EMSAs) using biotin-conjugated hARE1 and hARE2 and observed that nuclear proteins showed comparable binding activities to hARE1-Biotin, hARE2-Biotin, and Nrf2 standard-binding ARE (sARE-Biotin) (Fig. 6c, d), which was further confirmed by ChIP assays (Supplementary Fig. 6a, b). Consistent with these results, tBHQ treatment also increased the expression levels of Nestin and Nrf2-downstream genes at the mRNA and protein levels (Fig. 6e, f). These results indicate that Nrf2 increases Nestin expression by binding to the Nestin promoter. Moreover, when we established Nrf2-knockdown cells and explored their expression of Nestin, our qPCR, Western blot analysis and immunofluorescent staining results showed that Nrf2 knockdown markedly suppressed the expression of Nestin and its downstream genes (Fig. 6g–i). Taken together, these results demonstrate that Nestin is a target gene of Nrf2, and that Nrf2 promotes the transcription and expression of Nestin via a positive-feedback loop.

**Nestin and Nrf2 cooperatively enhance antioxidant capacity**. To further explore the role of Nrf2 in the Nestin-mediated resistance to oxidative stress, we used siRNAs or overexpression plasmids transfection to alter the levels of Nrf2 and Nestin in A549 cells, and stimulated the cells with $H_2O_2$ at 48 h post-transfection. The overexpression of Nrf2 rescued the capacity of Nestin-knockdown cells to resist $H_2O_2$-induced toxicity (Fig. 7a, b), whereas the suppression of Nrf2 enhanced the sensitivity of Nestin-overexpressing cells to $H_2O_2$-induced toxicity (Fig. 7c, d). Nestin knockdown slightly reduced the ability of Nrf2 to protect cells from oxidative stress, whereas Nestin overexpression partially rescued the sensitivity of Nrf2-depleted cells to $H_2O_2$-induced toxicity. To examine how the interaction between Nestin and Nrf2 affects downstream antioxidant genes, we used ARE luciferase reporter assays to detect changes in transcription. The results revealed that the overexpression of Nrf2 prevented the downregulation of the ARE pathway caused by the deficiency of Nestin, and vice versa, whereas the overexpression of Nestin had only a slight effect on ARE luciferase activity when Nrf2 was deleted (Fig. 7e, f). Consistent with the aforementioned results, detection of the total antioxidant capacity (Fig. 7g, h) and Western blot analysis of HO-1 and NQO1 (Fig. 7i) showed that the antioxidant capacity was increased in Nestin-knockdown cells subjected to overexpression of Nrf2 and decreased in Nestin-overexpressing cells subjected to knockdown of Nrf2.

To examine the effects of Nestin and Nrf2 on NSCLC progression, we generated xenograft models by subcutaneously injecting transfected NSCLC cells into nude mice. As shown in Fig. 7j–l, downregulation of Nestin caused significant reductions of tumor growth and weight, whereas overexpression of Nrf2 could enhance the tumor growth of Nestin-knockdown cells. In addition, Nrf2 depletion inhibited the rapid tumor growth caused by Nestin upregulation, indicating that the ability of Nestin to promote NSCLC development depends on the Nrf2–ARE signaling pathway.

**Relationship between Nestin and antioxidant capacity in vivo**. In addition, we analyzed the overall survival (OS) and disease free survival (DFS) in the two lung cancer patients groups of 15% cutoff high and 15% cutoff low Nestin expression, which was based on The Cancer Genome Atlas (TCGA) databases. The results fully showed that high levels of Nestin expression was closely correlated with poor survival and could be used as a prognostic biomarker for patients with lung cancer (Supplementary Fig. 7a).

To further examine the clinical relevance of the relationship between Nestin and Nrf2 in NSCLC cells, we assessed the expression of Nestin and Nrf2 in 200 NSCLC specimens using IHC analysis. A correlation study found that regions with high levels of Nestin staining also showed strong Nrf2 staining density,

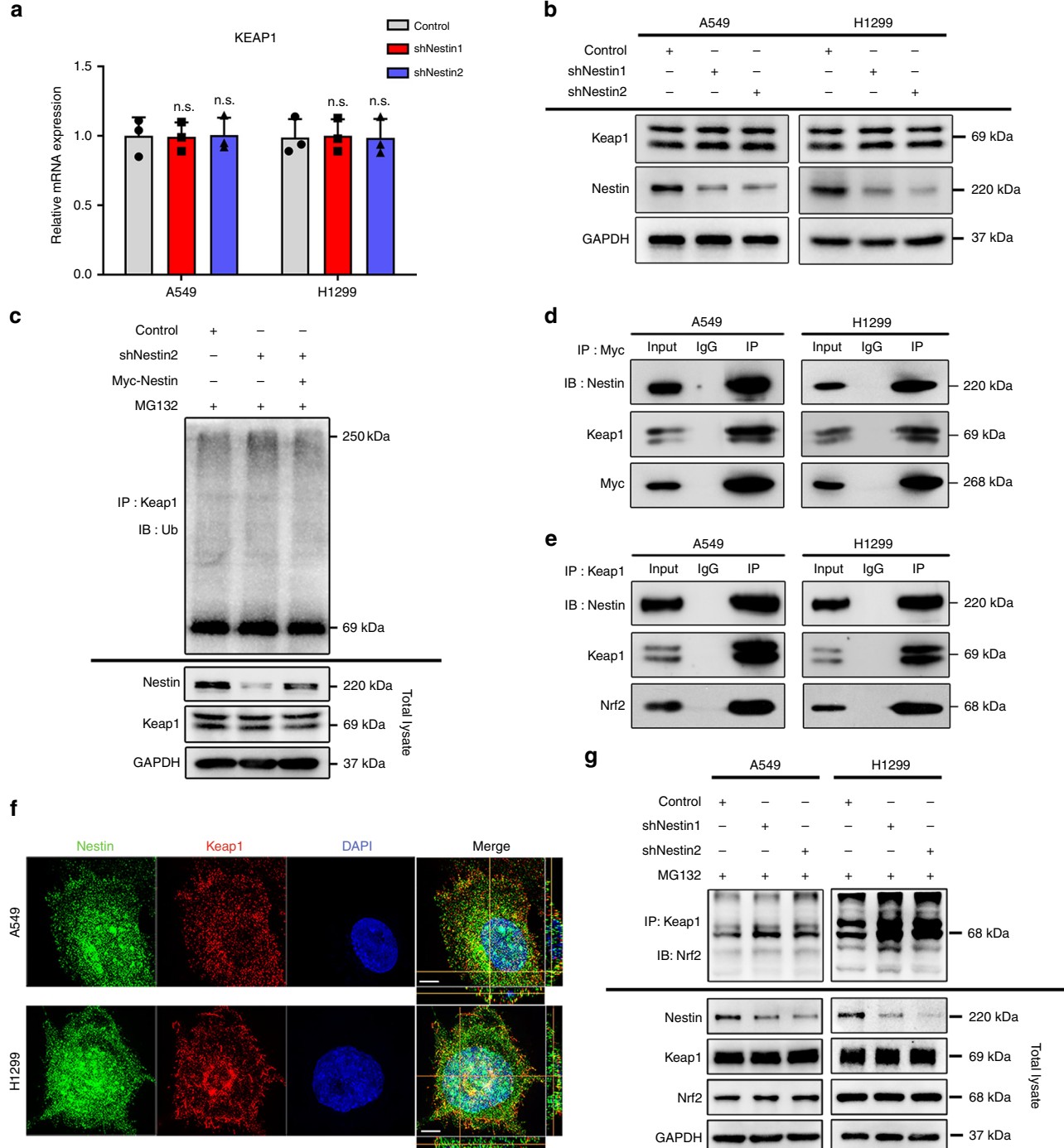

**Fig. 4** Nestin interferes with the Keap1-dependent ubiquitination of Nrf2 by competitively binding to Keap1. **a** qPCR analysis showing that knockdown of Nestin had no effect on Keap1 expression at the mRNA level. **b** Immunoblotting analysis showing that Nestin had no effect on Keap1 expression at the protein level. **c** Alteration of the Nestin levels had no influence on the ubiquitination of Keap1. Control or Nestin-knockdown cells transfected with or without a vector encoding Myc-Nestin were treated with 10 μM MG132 for 4 h and an in vivo ubiquitination assay was performed to determine the ubiquitination levels of Keap1. **d** Myc-Nestin plasmids were transfected into NSCLC cells, whole-cell lysates were immunoprecipitated with anti-Myc, and the precipitated proteins were blotted with the indicated antibodies. **e** Whole-cell lysates were immunoprecipitated with anti-Keap1 and the precipitated proteins were blotted with anti-Nestin, anti-Keap1, and anti-Nrf2. **f** The localizations of endogenous Keap1 and Nestin in NSCLC cells were determined by double-label indirect immunofluorescence with anti-Keap1 (red) and anti-Nestin (green) antibodies. The colocalization of Keap1 and Nestin is indicated by a yellow color in the merged images. Scale bar: 5 μm. **g** Nestin reduced the interaction between Nrf2 and Keap1. Control or Nestin-knockdown NSCLC cells were treated with 10 μM MG132 for 4 h. Cell lysates were immunoprecipitated with an anti-Keap1 antibody and blotted with an anti-Nrf2 antibody. N.S. represents no significant, Student's *t* test. Source data are available as a Source Data file

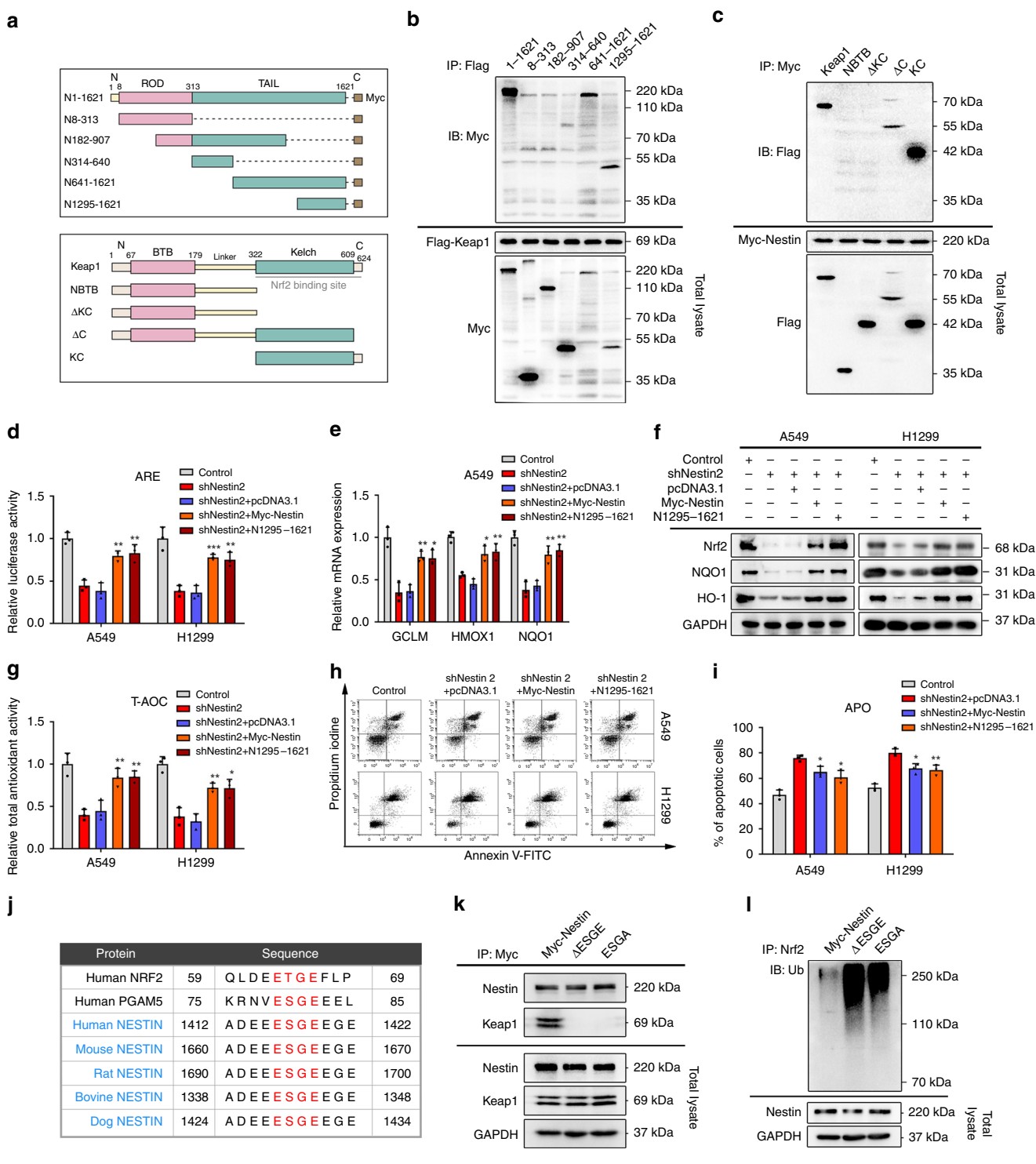

while those with low Nestin expression displayed weak Nrf2 signals, and that there was a statistically significant correlation between Nestin and Nrf2 expression (Fig. 8a). We also determined the mRNA levels of Nestin and Nrf2 target genes in 12 freshly collected clinical NSCLC samples. We found that the transcriptional level of Nestin was strongly correlated with those of HO-1, NQO1, GCLM, and GCLC (Fig. 8b). In addition, the percentage of tumors showing simultaneous upregulation of Nestin and Nrf2 was low in tumors of grade I and II but became markedly higher in those of grade III and was further elevated in grade IV tumors (Fig. 8c, d). Taken together,

these results show that the expression levels of Nestin and Nrf2 are valuable predictors of NSCLC malignancy.

Finally, in light of the antioxidant activity of Nestin, we further explored the ability of Nestin expression to protect NSCLC cells from chemotoxicity in response to 17-AAG, which is an HSP90 inhibitor that triggers oxidative stress[25]. Firstly, A549 or H1299 cell lines were treated with increasing concentrations of 17-AAG for 48 h, respectively. The results showed that 17-AAG significantly impaired the cell viability of NSCLC cell lines in a dose-dependent manner measured by CellTiter-Glo assays. The $IC_{50}$ of 17-AAG was about 85.2 nM for A549 cells and 71.8 nM for

**Fig. 5** The ESGE motif is essential for the ability of Nestin to interact with Keap1. **a** Schematic depiction of wild-type and deletion mutants of Myc-tagged Nestin and Flag-tagged Keap1. **b** A series of truncated Myc-tagged Nestin proteins were expressed with Flag-tagged Keap1 in HEK293FT cells. Immunoprecipitation was performed using Protein G beads and an anti-Flag antibody. **c** Truncated Flag-tagged Keap1 proteins were expressed with Myc-tagged Nestin in HEK293FT cells. **d** Nestin-knockdown NSCLC cells were transfected with empty pcDNA3.1, the same vector encoding Myc-Nestin or Nestin (N1295-1621). At 48 h post-transfection, the cells were transfected with the ARE luciferase reporter and subsequently assayed for luciferase activity. **e** Comparison of the mRNA expression levels of Nrf2-downstream genes in Nestin-knockdown NSCLC cells transfected with or without pcDNA3.1, Myc-Nestin or Nestin (N1295-1621) vectors. **f** Nestin-knockdown NSCLC cells were, respectively, transfected with pcDNA3.1, Myc-Nestin, or Nestin (N1295-1621) as indicated. The expression levels of Nrf2, NQO1, and HO-1 were analyzed via immunoblotting. **g** The total antioxidant activity (T-AOC) of the NSCLC cells described in **f** was assessed. **h** Nestin-knockdown NSCLC cells were transfected with pcDNA3.1, Myc-Nestin, or truncated Nestin (N1295-1621) plasmids for 72 h and then treated with 200 μM $H_2O_2$ for 4 h. Flow cytometry with Annexin V-FITC and PI was used to detect apoptosis. **i** Statistical analysis of the total apoptosis rates in the NSCLC cells described in **h**. **j** Highlighted sequence alignment of the putative Keap1-binding motif in Nestin from different species and those previously reported in Nrf2 and PGAM5. **k** NSCLC cells were transfected with plasmids encoding Myc-Nestin, ΔESGE, or ESGA mutants. Immunoprecipitation was performed using Protein G beads and an anti-Myc antibody. **l** NSCLC cells were transfected with plasmids encoding Myc-Nestin, ΔESGE, or ESGA mutants. Lysates were denatured, immunoprecipitated with anti-Nrf2, and blotted with anti-ubiquitin. Data are presented as the means ± SD of three independent experiments. *$P < 0.05$, **$P < 0.01$, and ***$P < 0.001$, Student's $t$ test. Source data are available as a Source Data file

H1299 cells (Supplementary Fig. 8a, b). Moreover, Nestin knockdown increased the sensitivity of NSCLC cells to 17-AAG, while Nestin overexpression increased the viability of 17-AAG-treated tumor cells (Supplementary Fig. 8c, d). In addition, our bioluminescent imaging demonstrated that Nestin knock-down rendered NSCLC cells more sensitive to 17-AAG in a subcutaneous tumor model in vivo, whereas the cytotoxicity of 17-AAG was attenuated in NSCLC tumors overexpressing Nestin (Fig. 8e, f). Consistent with these findings, tumor-weight assay indicated that Nestin expression could protect the NSCLC cells from chemotherapy-induced cell death (Fig. 8g). Collectively, these results show that chemotherapy via introduction of oxidative stress combined with Nestin silencing might be an effective treatment for NSCLC tumors.

## Discussion
Modulation of intracellular oxidative stress is now considered to be an effective anticancer therapy. Here, we show that Nestin, which is overexpressed in NCLSC, competitively binds to Keap1 via its Kelch domain and stabilizes Nrf2 by preventing its ubiquitin–proteasome-mediated degradation, thereby promoting the nuclear translocation of Nrf2 and increasing the antioxidant capacity. We also show that Nestin expression is downregulated upon knockdown of Nrf2. Taken together, our results reveal that Nestin and Nrf2 are involved in a positive-feedback loop that enables them to mediate the antioxidant responses and maintain cellular redox homeostasis (Fig. 8h).

In addition to be a common marker of multipotent stem cells[26–28], Nestin is also widely upregulated under conditions of tissue injury and cancer development[29,30]. Various studies have shed light on the molecular mechanisms underlying the contributions of Nestin to tumor progression in vivo and in vitro. For example, Nestin-positive progenitor cells in the cerebellum exhibit more efficient tumor cell transformation and severe genomic instability[31]. Nestin-expressing progenitor-like cells dedifferentiated from mature hepatocytes can develop into hepatocellular carcinomas or cholangiocarcinomas[32]. Moreover, Li et al. showed that Nestin binds Gli3, which is a transcription factor of the hedgehog pathway, to mediate the development of medulloblastomas[33]. Here, we report that Nestin plays an important role in maintaining the redox balance of NSCLC cells and can initiate intracellular responses to oxidative stress. As Nestin expression is tightly correlated to tumor malignancy, elucidating the underlying mechanisms through which Nestin is regulated and exerts its antioxidant function may suggest therapeutic targets in efforts to abrogate tumor growth and restore chemosensitivity.

Nrf2 acts as a key regulator of the expression of cytoprotective proteins and is regulated by a finely tuned control system. Homodimeric Keap1, which is a substrate adaptor protein for E3-ubiquitin ligase, targets Nrf2 for ubiquitination and degradation in the absence of oxidative stress. Therefore, Nrf2 protein levels may be affected by either disruption of Keap1–Nrf2 complex stability or reduced expression of Keap1. Several regulators have been shown to positively or negatively influence the Keap1–Nrf2 complex, leading to accelerated decay or consistent activation of Nrf2, respectively. Jain et al. and Yang et al. reported that p62/SQSTM1 and Gankyrin, respectively, could bind to Keap1 and protect Nrf2 from degradation[34,35]. Moreover, p21[36] and CDK20[37] were reportedly contribute to the upstream regulation of the Nrf2–Keap1–ARE pathway. Consistent with these findings, we herein show that Nestin can facilitate the stabilization of Nrf2 by competitively binding with Keap1 via its Kelch domain. In addition, Keap1 ubiquitination/deubiquitination post-translationally regulates the expression of Keap1 and is another important modulator of the Nrf2-dependent antioxidant response[12]. Villeneuve et al. demonstrated that USP15, which specifically deubiquitinates Keap1, promotes the stability of the Keap1–Nrf2 complex and the degradation of Nrf2[38]. Here, we show that Nestin knockdown has little effect on the expression and ubiquitination of Keap1, indicating that Nestin primarily influences the stability of Nrf2 by competitively binding with Keap1.

Accumulating evidence indicates that two separate sequences, the DLG and E(S/T)GE motifs, are responsible for binding to Keap1, with the E(S/T)GE motif showing a relatively higher binding affinity. Therefore, there are two distinct conformations of the interaction between Keap1 and Nrf2: the open and closed states. In the open state the ETGE motif of one Nrf2 interacts with a single molecule of Keap1, while in the closed state both the DLG and ETGE motifs of Nrf2 are bound to a Keap1 dimer. As oxidants inhibit the function of Keap1 rather than its binding, the tight-binding ETGE site may remain firmly attached to Keap1 under oxidative conditions, preserving the Keap1–Nrf2 association[39,40]. Recent reports have shown that some antioxidant proteins can compete with Nrf2 for the binding of Keap1 by inhibiting the proper binding of the DLG or E(S/T)GE motifs. For example, the p62/SQSTM1 protein binds to Keap1 by the linear sequence, STGE, which resembles the ETGE sequence that Nrf2 uses to interact with the Kelch domain of Keap1. Moreover, Ge et al. found that iASPP contains a DLT motif which resembles DLG in Nrf2, but without ETGE-like motif, implying that iASPP may bind to Keap1 in manner similar to that of Nrf2[14]. For most of the previously studied proteins, the binding activity toward Keap1 depends on a single motif. It was thus interesting that we

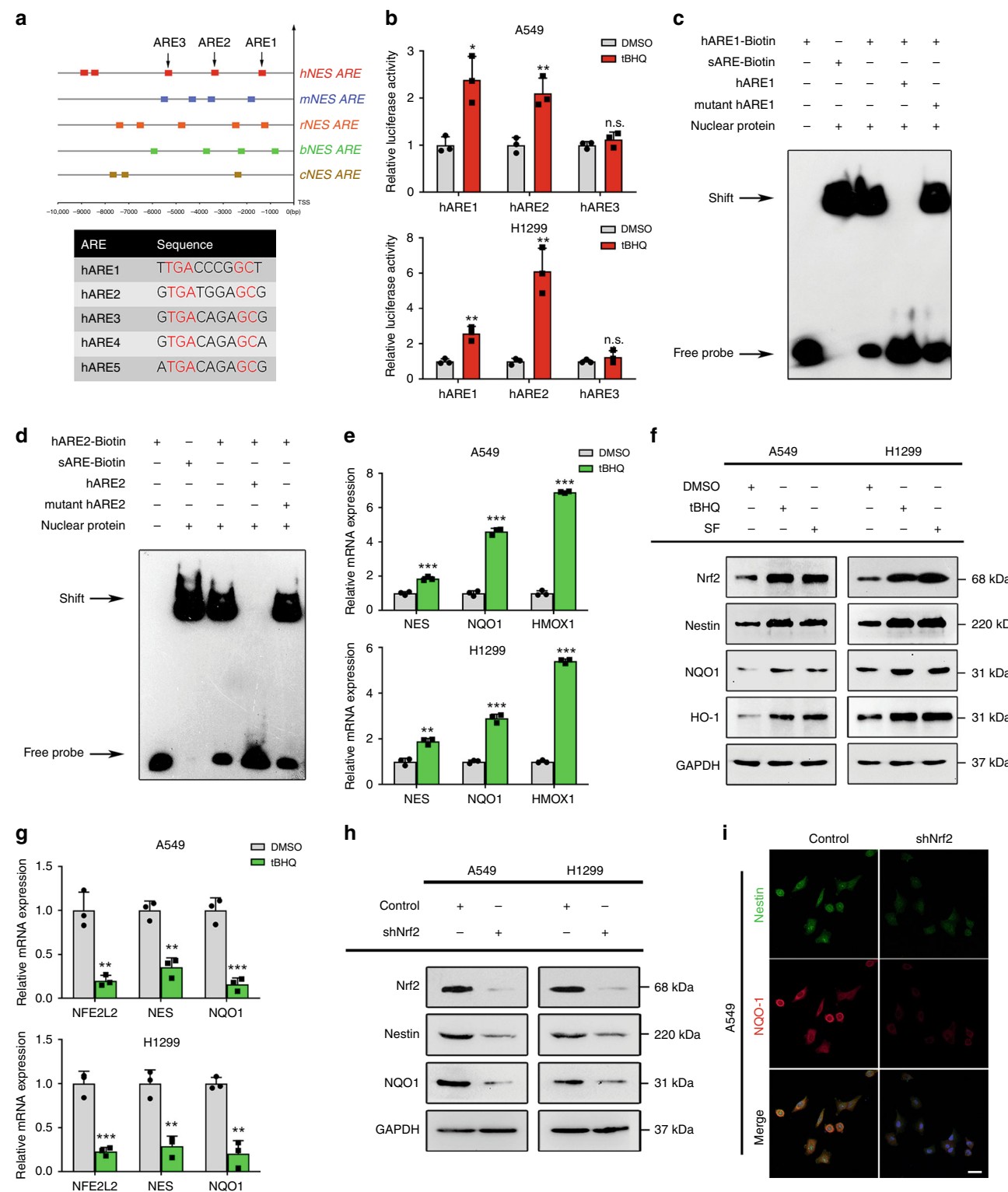

identified both DLG (N1371-1373) and ESGE motifs in the C-terminal tail domain of Nestin through sequence analysis. In the present work, we focused on the ESGE motif of Nestin and found that this high-affinity motif was responsible for the interaction with Keap1, and that deletion or missense mutation of the ESGE motif in Nestin reversed its ability to release Nrf2 from its inactivating complex, thereby increasing the proteasomal degradation of Nrf2. Future work is warranted to examine whether the DLG motif of Nestin interacts with Keap1. Given that Nestin has

both DLG and ESGE motifs, we speculate that Nestin, like Nrf2, may play pivotal roles in cellular adaptions to various stimuli and act as a master regulator to induce a diverse battery of genes with cytoprotective actions. Further studies are clearly needed to elucidate the detailed biological features of Nestin.

On the other hand, Masayuki et al. previously found that somatic mutation and gene variation of *KEAP1* is a common event in lung cancers and cancer-derived cell lines[41]. Moreover, Shyam et al. revealed that *KEAP1* had homozygous mutations in

**Fig. 6** Nrf2 promotes Nestin transcription in a positive-feedback model. **a** Schematic representation (upper panel) of the Nestin promoters from humans (*Homo sapiens*, hNES), mice (*Mus musculus*, mNES), rats (*Rattus norvegicus*, rNES), cattle (*Bos taurus*, bNES), and dog (*Canis lupus familiaris*, cNES). Different symbols represent the Nrf2-binding (ARE) sites of different species. TSS, transcriptional start site. **b** tBHQ increased the binding of Nrf2 to the AREs of the Nestin promoter. A549 cells and H1299 cells were transiently transfected with a luciferase reporter driven by the AREs of the Nestin promoter (hARE1, hARE2, or hARE3), and then were left untreated or treated with 100 μM tBHQ for 4 h. **c, d** The bindings of hARE1-biotin, hARE2-biotin, and sARE-biotin to nuclear extracts from NSCLC cells exposed to laminar shear stress were analyzed by EMSA. The binding specificity was tested in competition experiments using an excess of unlabeled intact oligonucleotides (hARE1 or hARE2 and mutant hARE1 or mutant hARE2). **e** qPCR showing that treatment with tBHQ (100 μM, 4 h) upregulated the transcriptional levels of Nestin and Nrf2-downstream genes in both A549 and H1299 cells. **f** Treatment with tBHQ (100 μM, 4 h) or SF (20 μM, 16 h) upregulated the protein levels of Nestin, Nrf2, and Nrf2-downstream genes (NQO1 and HO-1) in A549 and H1299 cells. **g** NSCLC cells were transfected with control or shNrf2 plasmids. qPCR analysis showed that Nrf2 knockdown reduced the mRNA levels of Nestin and an Nrf2-downstream gene (NQO1) in A549 cells and H1299 cells. **h** NSCLC cells were transfected with control and shNrf2 plasmids. Immunoblotting analysis demonstrated that Nrf2 knockdown reduced the protein levels of Nestin and NQO1. **i** NSCLC cells with or without Nrf2 knockdown were labeled with anti-NQO1 (red), anti-Nestin (green), and DAPI (blue). Scale bar: 40 μm. Data are presented as the means ± SD of three independent experiments. *$P < 0.05$, **$P < 0.01$ and ***$P < 0.001$, Student's *t* test. Source data are available as a Source Data file

some lung cancer cell lines, such as A549, H460, and H1435 cells[42], suggesting that mutation *Keap1* may weaken its ability to modulate the activity of Nrf2. We then compared the interaction between WT/mutation Keap1 and Nestin, as well as Nrf2, using A549 (Keap1 G333C) and H1299 (Keap1 WT). The results showed that although Keap1 bound more ubiquitined-Nrf2 in H1299 than in A549, Keap1 still had the ability to bind with Nestin or Nrf2 in A549 cell line (Fig. 4g), indicating that mutations (G333C) in first Kelch domain of Keap1 might not result in complete dissociation with DLG and ETGE motifs in Nestin and Nrf2.

Despite the aforementioned evidence for the regulation of Nrf2 via Keap1, there are several clues showing that phosphorylation of Nrf2 induced translocation into nucleus independently of Keap1, such as by PKC, CK2, Cdk5, and so on[43–45]. Interestingly, Sahlgren et al. demonstrated the interaction between Nestin and Cdk5 in neuronal precursor cells under oxidative stress[20]. Similarly, in our previous studies, we found that downregulation of Nestin induced the activation of Cdk5 in NSCLC cells[46]. If so, Nrf2 would be phosphorylated and translocated into nucleus, subsequently enhancing antioxidant capacity, which seemed contradictory to our recent findings. In addition, we found that the phosphorylation levels of Nrf2 were decreased after Nestin knockdown. Thus, it seems that Cdk5 might not be involved in the signaling pathway of Nestin on protecting Nrf2 from Keap1-mediated degradation.

The expression of Nestin is a tightly regulated process. Recent studies have revealed several transcription factor-dependent and epigenetics-dependent mechanisms were responsible for regulating Nestin expression in tumorigenesis, tissue repair, and embryonic development. For example, Gomes et al. reported that Nestin expression is subject to negative epigenetic control by TET2 via the binding of 5-hydroxymethylcytosine (5-hmC) at the 3′ untranslated region of the Nestin gene, and that this down-regulation correlates with the growth of invasive melanoma[47]. Moreover, loss of p53 relieves the restriction on Nestin expression in an Sp1/3 transcription factor-dependent manner and facilitates tumor initiation in liver cancer[32]. In addition to its effects on common antioxidant genes, Nrf2 has been reported to act as a transcription factor for some oncogenes, such as Klf9[48], p62/SQSTM1[34], and ATF3[49]. Therefore, we hypothesized that the apparent oncogene, Nestin, could also be regulated by Nrf2 activation. Interestingly, when we searched the Nestin gene promoter for transcription factor-binding sequences, we identified several conserved AREs near the transcriptional start site of the Nestin gene. Moreover, we herein show that Nestin is upregulated through the Nrf2–ARE signaling pathway. Meanwhile, knock-down of Nrf2 reduced the expression of Nestin, suggesting that Nestin was a downstream target gene of Nrf2. In brief, these findings expand our understanding of the mechanism through which Nrf2 activation modulates Nestin. Notably, we demonstrate that there is a positive-feedback loop between Nestin and Nrf2, and that it is responsible for mediating the antioxidant responses and maintaining cellular redox homeostasis in lung cancer.

Taken together, our data reveal the molecular basis for the positive-feedback loop between Nestin and Nrf2, which critically contributes to mediating antioxidant responses and maintaining cellular redox homeostasis. Our findings suggest that the Nestin–Nrf2 signaling pathway and antioxidant defenses could be targeted as promising therapeutic approaches for cancer treatment.

## Methods

**Human subjects and samples.** Two hundred Paraffin-embedded archived specimens that had been histopathologically diagnosed as NSCLC were obtained from the First Affiliated Hospital of Sun Yat-sen University, China. Donor informed consent had been previously obtained from patients, and approval was obtained from the Committees for Ethical Review of Research involving Human Subjects of Sun Yat-Sen University, China. Specimens collected for gene expression analysis were stored at liquid nitrogen immediately after surgery.

**Mice and tumor models.** For mouse xenograft models, the BALB/c nude mice were purchased from Beijing Vital River Laboratory, housed under standard specific-pathogen-free (SPF) conditions, and randomly allocated into groups receiving cell line injections. The Dox (Doxycycline)-induced xenograft model was created by subcutaneously implanting $5 \times 10^6$ inducible Nestin-knockdown NSCLC cells into the right flanks of 6-week-old male BALB/c nude mice ($n = 3$ mice/group). Tumors were allowed to initiate/grow for 2 weeks, and the mice were than treated with or without Dox (625 mg/kg food) in their diets for 3 weeks. Tumors were collected for immunohistochemistry or immunofluorescence. To generate shRNA xenograft models, we first transfected NSCLC cells with shRNA vectors specifically targeting Nestin or Nrf2, and then infected the cells with Nrf2 or Nestin-expressing lentiviruses, respectively. The target sequences of shRNAs against Nestin and Nrf2 were listed in Supplementary Table 2. Approximately $5 \times 10^6$ A549-control, A549-ovNestin, A549-shNestin, A549-shNestinovNrf2, or A549-ovNestinshNrf2 cells were suspended in 0.1 ml PBS and injected subcutaneously into the right flanks of 8-week-old male BALB/c nude mice ($n = 4$ mice/group), which were then observed for tumor development every 2 days. For 17-AAG-treated xenograft model, we subcutaneously injected $5 \times 10^6$ A549-luciferase, A549-ovNestin-luciferase, or A549-shNestin-luciferase cells into the right flanks of nude mice ($n = 4$ mice/group). Three groups including the ones injected with A549-luciferase, A549-ovNestin-luciferase, or A549-shNestin-luciferase cells were treated with 25 mg/kg 17-AAG (Selleck), three times per week for 3 weeks. One group of mice injected with A549-luciferase cells were treated with PBS for control. The bioluminescence imaging (BLI) was applied to detect the signal of luciferase. All animal procedures were approved by the Ethical Committee of Sun Yat-sen University and were conducted in accordance with the animal care guidelines of the National Institutes of Health (NIH) and with the ethical guidelines. No method of blinding was used.

**Cell culture experiments.** The A549, H1299 and HEK293FT cell lines were obtained from the American Type Culture Collection (ATCC). Cells were cultured in Dulbecco's modified Eagle's medium (DMEM; Hyclone) supplemented with 10% vol/vol) fetal bovine serum (FBS; Invitrogen), 100 IU/ml penicillin (Hyclone), and 100 μg/ml streptomycin (Hyclone), in an atmosphere of 95% air and 5% $CO_2$. All cell lines have been tested negative for mycoplasma contamination.

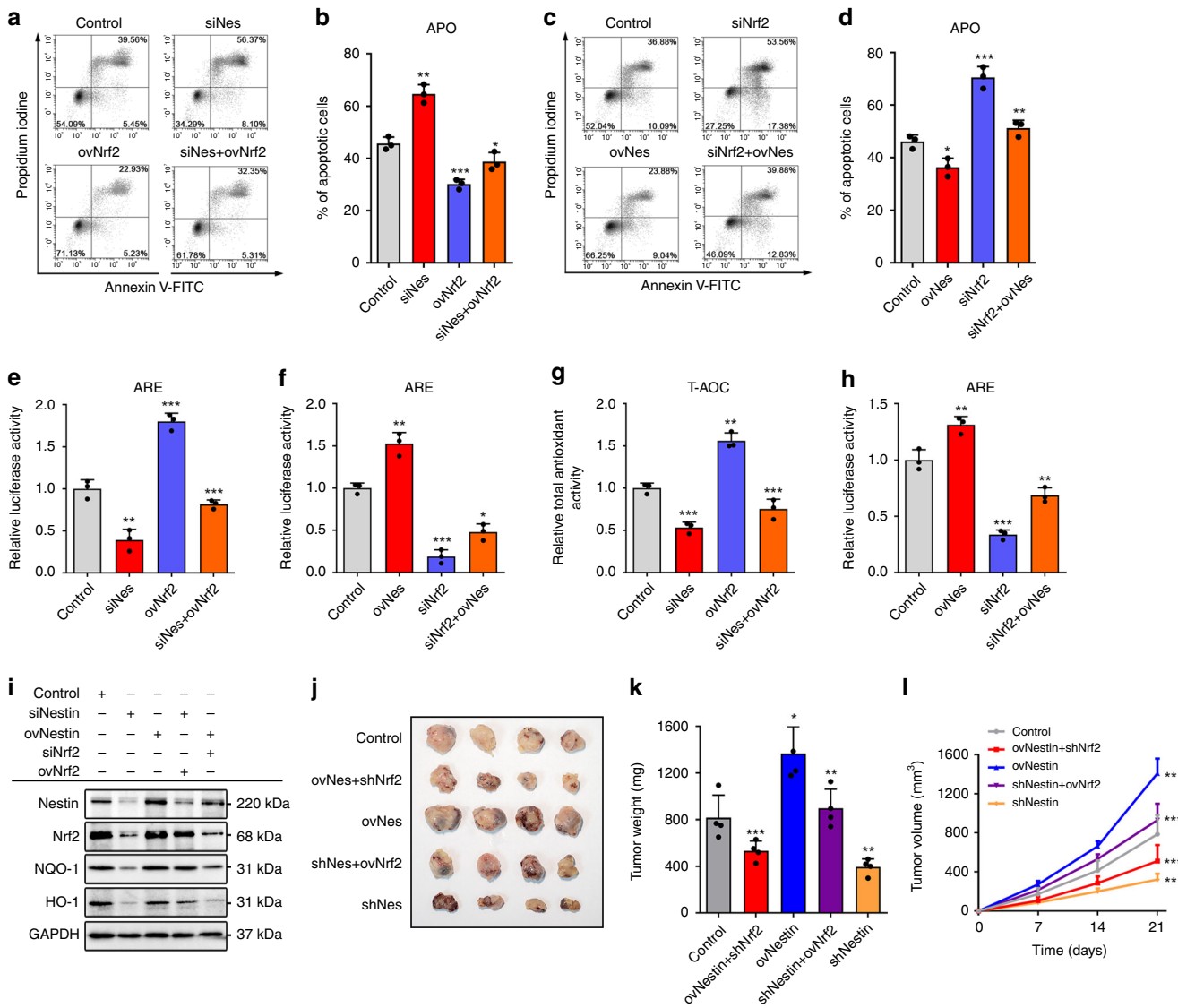

**Fig. 7** Nestin and Nrf2 cooperatively contribute to the antioxidant stress capacity of NSCLC cells. **a** Flow cytometric analysis performed with Annexin V-FITC and PI was used to detect apoptosis of A549-con, A549-siNestin, A549-ovNrf2, and A549-siNestinovNrf2 cells, respectively. **b** Statistical analysis of the total apoptosis rate in A549 cells described in **a**. **c** Flow cytometric analysis with Annexin V-FITC and PI was used to detect apoptosis of A549-con, A549-siNrf2, A549-ovNestin, and A549-siNrf2ovNestin cells, respectively. **d** Statistical analysis of the total apoptosis rate in A549 cells described in **c**. **e** A549 cells were transiently treated with siNestin RNA and/or ovNrf2 plasmids, transiently transfected with an ARE-driven luciferase reporter gene construct, harvested, and assayed for luciferase activity. **f** A549 cells were transiently treated with siNrf2 RNA and/or ovNestin plasmids, transiently transfected with an ARE luciferase reporter gene construct, harvested, and assayed for luciferase activity. **g** Total antioxidant activity (T-AOC) was assayed in A549 cells that had been transfected with siNestin RNA and/or ovNrf2 plasmids. **h** Total antioxidant activity (T-AOC) was assayed in A549 cells that had been transfected with siNrf2 RNA and/or ovNestin plasmids. **i** Western blotting analysis was performed to evaluate the levels of NQO1 and HO-1 in A549 cells transfected with siNestin, ovNestin, siNestinovNrf2, or siNrf2ovNestin. **j, k** Nude mice were randomized into five groups ($n = 4$ per group) and subcutaneously injected with A549 cells that had been transfected with control, shNestin, ovNestin, shNestinovNrf2, or ovNestinshNrf2 plasmids. Tumors formed in nude mice were collected 21 days after grafting, and the tumor weight were measured. **l** The tumor volume (growth rate) was measured over time. Data are presented as the means ± SD of at least three independent experiments. *$P < 0.05$, **$P < 0.01$, and ***$P < 0.001$, Student's $t$ test. Source data are available as a Source Data file

**Construction of vectors**. The truncation mutants of Keap1 were generous gifts from B. Xia (University of Medicine and Dentistry of New Jersey, New Brunswick, NJ)[6]. A series of Nestin truncation mutants including Nestin (1-1621), Nestin (8-313), Nestin (314-640), Nestin (641-1621), Nestin (1295-1621) were constructed in our laboratory. The encoding cDNAs were PCR amplified and subcloned into the pcDNA3.1-Myc vector (Invitrogen) using appropriate restriction enzyme digests. The detailed information of the plasmids was listed in Supplementary Table 3. Two Nestin mutation plasmids, Nestin-ΔESGE (deletion mutant) and Nestin-ESGA (harboring a site-specific mutation that 1417E was replaced by 1417A within the Nestin open-reading frame) were constructed by PCR and fused into the BamHI/ XhoI sites of pcDNA3.1-Myc. For knockdown of Nestin and Nrf2 expression,

retrovirus vectors (pSM2) encoding shRNAs were purchased from Open Biosystems (Huntsville, AL, USA). Myc-Nestin and Flag-Nrf2 overexpression vectors were constructed using Invitrogen's Gateway System.

**RNAi transfection**. ShRNA transfections were performed using the MegaTran 1.0 Transfection Reagent (OriGene) according to the manufacturer's instructions. The lentiviruses were used to infect NSCLC cells with Polybrene (8 μg/ml) for 4 h. The original medium was replaced with fresh medium 12 h later. The siRNAs, siNrf2, and siNestin, were purchased from Ribobio, and their encoding vectors were

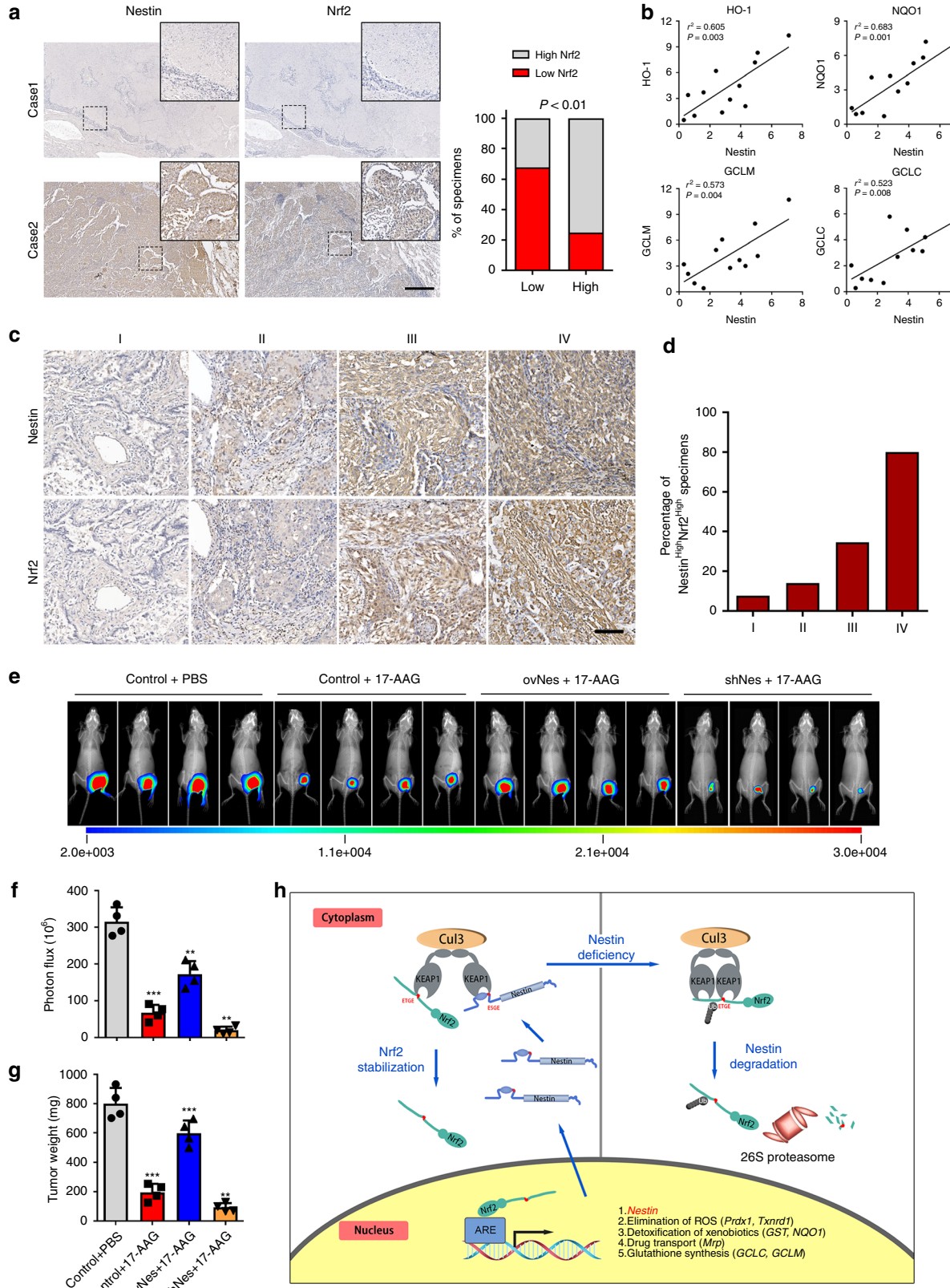

transfected into NSCLC cells using the Lipofectamine RNAiMAX Transfection Reagent (Invitrogen).

**Western blotting**. For immunoblotting, cell lysates were prepared using RIPA buffer (Millipore), supplemented with protease inhibitor cocktail (Roche). Then cell lysates were centrifuged at $10,000 \times g$ for 10 min at 4 °C to remove cell debris. After total protein concentration was assessed BCA Protein Assay Kit (Thermo),

20 µg of protein was denatured and resolved by SDS/PAGE, then transferred to PVDF membranes (Millipore). The target proteins were immunoblotted with the specific antibodies. The antibodies used can be found in the Supplementary Table 4. Chemiluminescent substrate (Millipore) was used for detecting the signaling intensity. For cytoplasm and nuclear protein separation, the Subcellular Protein Fractionation Kit (Thermo) was applied for distinguishing the cytoplasm and nuclear protein. The collected proteins were submitted for immunoblotting.

**Fig. 8** Nestin/Nrf2 overexpression in NSCLC correlates with 17-AAG resistance. **a** IHC was performed to examine the relationship between Nestin expression and Nrf2 expression in 200 primary NSCLC specimens. The results obtained from two representative cases are shown (left). Percentages of specimens showing low or high Nestin expression relative to the levels of Nrf2 staining (right). Scale bar: 500 μm. **b** Nestin expression positively correlated with the expression levels of Nrf2-target genes (HO-1, NQO1, GCLM, and GCLC) in NSCLC specimens. The mRNA levels of Nestin and Nrf2-target genes were detected by qRT-PCR, and the correlations between the mRNA levels of Nestin and those of various antioxidant enzymes were evaluated using Pearson's correlation coefficients ($n = 12$). **c** The expression levels of Nestin and Nrf2 were assessed in NSCLC tumors of grades I–IV. Scale bar: 100 μm. **d** The percentage of tumors showing simultaneous upregulation of Nestin and Nrf2 gradually increased from grade I to grade IV. **e** A549 cells were transfected with siNestin RNA or ovNestin plasmids. After 5 days, the cells ($5 \times 10^6$ cells) were subcutaneously injected into nude mice ($n = 4$ per group), which were thereafter treated with 17-AAG three times a week. Tumor growth was examined by bioluminescent imaging on day 21. **f** Bioluminescent density was quantified. **g** Tumor weight on day 21 after mice were xenografted with NSCLC cells transfected with Control, shNestin, or ovNestin plasmids, with or without 17-AAG treatment. **h** Proposed model of the relationship between Nestin and Nrf2 in NSCLC cells. Data are presented as the means ± SD of at least three independent experiments. *$P < 0.05$, **$P < 0.01$, and ***$P < 0.001$, Student's $t$ test. Source data are available as a Source Data file

**Co-immunoprecipitation**. To detect the endogenous interaction, cells were transfected with or without indicated plasmids for different experimental purpose and lysed in IP lysis buffer (Thermo) containing a protease inhibitor cocktail (Roche). The cell extracts were purified by centrifugation at $10,000 \times g$ for 10 min at 4 °C. Then the supernatants were incubated with indicated antibodies or IgG control derived from the same species as the indicated antibody overnight at 4 °C, followed by incubation with Protein G magnetic beads (Thermo) for 2 h at 4 °C. The reaction mixtures were washed three times with IP lysis buffer supplemented with protease inhibitors, and harvested by centrifugation. After the immunoprecipitated proteins were denatured, the direct interactions between proteins were analyzed by immunoblotting performed according to standard protocols. The antibodies used for immunoprecipitation can be found in Supplementary Table 4.

**Ubiquitination assay**. To detect the ubiquitination levels of endogenous Nrf2 or Keap1, cells were transfected with or without indicated plasmids for different experimental purpose and treated with 10 μM MG132 (Sigma) for 4 h to block proteasomal degradation. The cells were lysed and immunoprecipitated for 4 h at 4 °C with Protein G Magnetic beads (Thermo) loaded or bound with anti-Keap1 (Santa Cruz), anti-Nrf2 (Abcam), or anti-Myc (Santa Cruz) antibodies according to immunoprecipitation assay described above. The immunoprecipitated proteins were subjected to immunoblotting analysis with antibody against Ubiquitin (Santa Cruz).

**RNA extraction, cDNA synthesis, and real-time quantitative PCR**. Total RNA was extracted from H1299, A549 cells and fresh-frozen tumor specimens using the TRIzol Reagent (Invitrogen) according to the manufacturer's instructions. Equal amounts of mRNA were used to generate cDNAs with a Revert Aid First Strand cDNA Synthesis Kit (Thermo), and the generated cDNAs were used for real-time quantitative-PCR (qPCR). qPCR was performed using a 480 SYBR Green I Master kit (Roche) and a LightCycler480 Detection System (Roche). The primer sequences used for real-time PCR are listed in Supplementary Table 5. The data were normalized to GAPDH and were expressed as relative mRNA levels.

**ChIP assay**. ChIP assay was performed according to the manufacturer's instructions of the SimpleChIP enzymatic ChIP kit (Cell Signaling Technology). Chemical crosslinking of DNA–proteins was carried out using 1% formaldehyde for 10 min at room temperature. The crosslinking was quenched by addition of glycine (0.125 M) for 5 min at room temperature and followed by two washes with ice-cold PBS. Cells were scraped into PBS containing Protease Inhibitor Cocktail (200X) provided by kit. The cell suspension was centrifuged and the pellet was mixed by inverting the tube every 3 min in buffer A + DTT + PIC followed by incubation on ice for 10 min. The pellet (containing nuclei) was dissolved in 1.0 ml buffer B + DTT + 5 ml of micrococcal nuclease and incubated for 20 min at 37 °C with frequent mixing to digest DNA to a length of ~150–900 bps. The lysates were immunoprecipitated using ChIP-grade Nrf2 antibody or normal rabbit IgG overnight at 4 °C with rotation and followed by ChIP-grade protein G magnetic beads and incubation for 2 h at 4 °C with rotation. The magnetic beads were washed using buffers supplied with the kit. The eluted DNA was purified and analyzed by qPCR to determine the binding of Nrf2 to the *NESTIN* promoter. The primer sequences used for real-time PCR are listed in Supplementary Table 5.

**Cell viability**. Cell viability based on ATP measurement was assessed using Cell-Titer Luminescent Cell Viability Assay (Promega). Briefly, NSCLCs cells were seeded in a 96-well plate at a density of $2 \times 10^4$/well and treated with gradient concentration of 17-AAG. After incubation, 100 μL staining solution (CellTiter-Glo reagent) was added to each well and mixed for 2 min on an orbital shaker to induce cell lysis. Cells were incubated at room temperature for 10 min to stabilize the luminescence signal, which was recorded using the microplate reader. The experiment was run in triplicate. The plate was incubated for 10 min and the luminescent signal was recorded using Infinite® F200 pro microplate reader (Tecan).

**Apoptosis assay and flow cytometry**. After indicated treatments, NSCLC cells were incubated with 200 μM $H_2O_2$ for 6 h. Subsequently, both suspended and attached cells were collected gently in 100 μL and incubated with 5 μL FITC-conjugated Annexin V and 5 μL PI (Vazyme Biotech) for 10 min at room temperature in dark. Samples were run on a CytoFLEX (Beckman Coulter, USA) and the data were analyzed using the Flow Jo software (Tree Star Inc., Ashland, Oregon). All the cells were gated and at least 20,000 cells were collected for each sample.

**Detection of antioxidant capacity**. The GSH content, SOD activity, catalase activity, and total antioxidant capacity were measured using a GSH-Glo Glutathione Assay kit (Promega), a SOD Assay kit (Sigma-Aldrich), a Catalase Activity Assay kit (Biovision), and a Total Antioxidant Capacity Assay kit (Beyotime Biotechnology), respectively, according to the manufacturers' recommendations.

**Immunocytochemistry staining**. For immunocytochemistry staining, cells grown on cover slips were fixed in 3.7% formaldehyde, permeabilized in 0.2% Triton X-100, incubated with an appropriate primary antibody overnight at 4 °C, and then treated with a secondary antibody for 1 h in the dark at room temperature (the utilized antibodies are listed in Supplementary Table 4). Nuclei were stained with DAPI 5 min to enable quantification of the total nuclear intensity for all indicated targets. Images were acquired at room temperature using an LSM780 confocal microscope (Zeiss) and an A1R N-SIM (Nikon).

**Immunohistochemistry (IHC)**. For IHC staining, the tumor samples were fixed with 4% neutral-buffered paraformaldehyde and embedded in paraffin for sectioning. After de-paraffinization, dehydrated and antigen retrieval steps according to standard procedures, 5-μm paraffin sections were used for immunostaining. The indicated antibodies are listed in Supplementary Table 4. Then, two independent investigators, who were blinded to group-identifying information, scored the staining signaling in IHC-stained sections as follows: (1) the proportion of tumor cells, where no positive tumor cells were represented as 0, <10% positive tumor cells were represented as 1, 10–50% positive tumor cells represented as 2, and >50% positive tumor cells were represented as 3; and (2) staining intensity, where no staining was indicated as 0, weak staining (light yellow) was indicated as 1, moderate staining (yellow-brown) was indicated as 2, and strong staining (brown) was indicated 3. The staining index was calculated as the proportion of positive tumor cells score × the staining intensity score. A staining index score ≥4 was taken as high-level expression of the protein, while a score <3 was indicated as low-level expression. Images were collected by AxioScan.Z1 (Zeiss) and analyzed using the Image J software.

**Electrophoretic mobility shift assay**. Double-stranded oligonucleotides containing the human Nestin gene ARE-like site with and without mutation were end-labeled with [c-32P] ATP using T4 polynucleotide kinase. DNA–protein interactions were detected by electrophoresis on non-denaturing 6% polyacrylamide gels in Tris borate–EDTA (TBE) buffer, followed by autoradiography. For competition experiments, a 200-fold molar excess of either unlabeled probe or a random 43-base oligonucleotide was included in the preincubation mixture at 25 °C before the addition of the labeled probe. For supershift analyses, appropriate antibodies against Nrf2 or an unrelated rabbit IgG antibody (Santa Cruz) were used. A reaction volume of 5 μL containing 8–10 μg nuclear extract was mixed with 2 μL of the appropriate antibody (4 μg), quickly treated with the labeled oligonucleotide probe, and then incubated at room temperature for 30 min. After the electrophoresis was performed, an image of the gel reveals the positions of the free and bound $^{32}$P-labeled DNA.

**Generation of Dox-inducible shRNA clones**. Tet-On 3G Inducible Expression System, which contains pTRE3G-IRES and pLVXTet3G vectors, was purchased from Clontech (Supplementary Table 3). EF1α-Tet-On 3G-bsd and pTRE3G-

shNestin-neo were constructed using multisite Gateway technology[21]. NSCLC cells were transfected with these two vectors, followed by selection with blasticidin and neomycin (Sigma-Aldrich). To induce shRNA expression, the surviving NSCLC cells were treated with Dox (600 ng/ml) for 72 h.

**Construction of reporter plasmids and luciferase assays**. The antioxidant response element (ARE) and the Nestin-ARE (ARE like site) were cloned into pGL3-basic luciferase reporter plasmid. NSCLC cells ($2.5 \times 10^4$ cells per well) were seeded in triplicate to 24-well plates (Corning). After incubation for 24 h, the cells with either Nestin plasmids or shNestin were transfected with 200 ng of ARE-luciferase-reported plasmids using the MegaTran 1.0 Transfection Reagent (Ori-Gene) according to the manufacturer's recommendations. As for the detection of Nestin-ARE reporters, NSCLC cells were transfected with 200 ng of Nestin-ARE1 or Nestin-ARE2 plasmids. And each of transfection was included the same amount of Renilla, which was used to standardize transfection efficiency. After that cells were then allowed to recover in medium containing 10% FBS for 24 h. Forty-eight hours post-transfection, firefly and renilla signals were measured using a Dual Luciferase Reporter Assay kit (Promega) and presented as the increase in activation over reporter alone. For the in vivo luciferase experiment, lentiviral vectors PHBLV-ZsGreen-fLUC purchased from Hanheng Biotechnology were transfected into the A549 cells and screened by FACS. $5 \times 10^6$ cells were then subcutaneously inserted into the right flanks of 6-week-old male BALB/c nude mice ($n = 4$ mice/group). The bioluminescence imaging (BLI) was applied to detect the signal of luciferase, which were reacted with substrate, D-luciferin (Goldbio), intraperitoneally injected into the mice.

**Statistical analysis**. All experiments were performed at least three times and data were expressed as means ± standard deviation (SD) unless otherwise specified. Comparisons between groups were performed using the Student's $t$-test before Gaussian distribution was assumed. The associations between Nrf2 and Nestin expression levels were analyzed using the $\chi^2$ test. Correlation analyses of gene expressions were performed using Pearson's correlation coefficients. GraphPad Prism 7 Software was used for statistical analysis. A two-sided $P$-value < 0.05 was considered to be statistically significant. The level of significance is indicated as *$P$ < 0.05, **$P$ < 0.01, and ***$P$ < 0.001.

**Reporting summary**. Further information on research design is available in the Nature Research Reporting Summary linked to this article.

## Data availability
The source data underlying Figs. 1–8 and Supplementary Figs. 1–3, 5, 6, and 8 are provided as a Source Data file. All the other data supporting the findings of this study are available from the corresponding authors upon reasonable request.

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

## Acknowledgements

This work was supported by the National Key Research and Development Program of China, Stem cell and Translational Research (2018YFA0107200, 2017YFA0103403, 2017YFA0103802), Strategic Priority Research Program of the Chinese Academy of Sciences (XDA16010103, XDA16020701), the National Natural Science Foundation of China (81425016, 81570593, 81730005, 31771616, 81802402, 81971372), the Natural Science Foundation of Guangdong Province (2017A030310237), the Key Research and Development Program of Guangdong Province (2016B030229002, 2017B020231001, 2019B020234001, 2019B020236002, 2019B020235002), Key Scientific and Technological Program of Guangzhou City (201803040011, 201704020223), the Fundamental Research Funds for the Central Universities (19ykpy158, 19ykyjs55, 19ykyjs56, 19ykyjs60) and China Postdoctoral Science Foundation (2017T100657).

## Author contributions

A.P.X., M.S.W., K.Z., J.C.W., Q.Y.L., and J.Y.C. designed the experiments and wrote the paper. Q.Y.L., Y.W., Y.Q., X.F.L., Y.N.Z., H.X.W., Y.Y.W., X.L., and Y.S. performed research; Q.Y.L., J.Y.C., Y.W., Y.N.Z., H.X.W., Y.Y.W., and Y.N.H. analyzed the data. J.C.W. and J.Y.C. collected clinical samples. Y.J.G., Q.K., and Y.Q. contributed reagents/analytic tools.

## Competing interests

The authors declare no competing interests.
