## [Peer Review File · Nature Communications]

Reviewers' comments:

Reviewer #1 (Remarks to the Author):

In this manuscript Wang et al., investigate the role of Nestin in modulation of the oxidative stress. The authors utilise a biochemical approach to demonstrate that Nestin competes with Nrf2 for direct binding with Keap1. The net result of this interaction is the escape of Nrf2 degradation via the proteasome, resulting in activation of the anti-oxidant transcriptional program. Moreover, the authors demonstrate that this process occurs through a feedback loop, as ARE elements were found to reside in the Nestin gene promoter and are essential for Nrf2-mediated transcriptional activation of Nestin.

There is immense interest in the mechanisms by which augmented Keap1-Nrf2 pathway activity promotes malignant transformation and therapeutic response in cancer (lung cancer). The authors demonstrate that Nestin is a novel Keap1-interacting protein, and provide evidence for the direct transcriptional regulation of Nestin by Nrf2.

Specific comments:

1. The title "Nestin regulates cellular redox homeostasis and drug resistance in lung cancer through the Keap1-Nrf2 feedback loop" does not faithfully reflect the aim findings of the paper. Further evidence would be required in order to state that Nestin plays a role in "drug resistance" in this setting.

2. Two lung adenocarcinoma cell lines were used for this study; A549 and H1299. A549 cells are KEAP1-mutant while H1299 cells are KEAP1-WT. Can the authors please comment on what effect this mutation may have on the subsequent findings of the paper? Particularly with respect to the ability of Nestin to "compete" with Nrf2 for Keap1 binding, as the Keap1-Nrf2 interaction in these cells is likely to be abnormally perturbed.

3. Figure 1 – Decreased expression of a panel of anti-oxidant genes are observed in both A549 and H1299 cells following Nestin knockdown. Are the expression changes shown in panel D and E statistically significant? The authors hypothesize that modulation of the anti-oxidant response may have therapeutic potential. Did the author's observe effects on tumour growth in experiments outlined in I-L? What cell line is represented in panel K? Images of single channel stains: NQO1, HO-1 and GCLM should also be included in panel L.

4. Figure 3 – The activity of Nrf2 is tightly controlled by its subcellular localisation. In the cytoplasm Nrf2 is degraded via the proteasome, through its interaction with KEAP1 and the CUL3 ubiquitin ligase complex. shNestin 2 was used to evaluate the cytoplasmic/nuclear localisation of Nrf2 in protein fractionation studies. The selection of shNestin 2 for these experiments is unclear, given that in Figure 2 Nrf2 protein expression was most markedly reduced (absent) upon Nestin KD with shNestin 2. Did the author's carry out fractionation experiments in A549 cells following KD of Nestin using shNestin 1?

5. Further clarification is required in relation to the IP/pull-down experiments shown in Figure 4. Are these experiments performed with endogenous KEAP1? Given that KEAP1 is mutated in A549 cells, how does this effect the results presented?

6. FL-Nestin should be included as an additional control in rescue experiments outlined in panels D-I.

7. Figure 6 – Panel A: Can a complete sequence alignment be included as supplementary information? Can the authors please comment on why only hARE1 and hARE2 were selected for functional assays? Direct binding of Nrf2 to the Nestin promoter is demonstrated using EMSAs. Do ChIP assays confirm these findings? Have the author's interrogated publicly available Nrf2 ChIP-seq datasets to assess whether Nestin also comes up as a direct target? Chorley et al., *Nucleic Acids Res* 2012; Malhotra et al., *Nucleic Acids Res* 2010.

8. Correlations between Nrf2 and Nestin expression have been assessed in a panel of 200 NSCLC

specimens. Have the authors correlated the expression of NQO1 (a robust biomarker of Nrf2 activity; Romero et al., Nature Medicine 2017; Best et al., Cell Metabolism 2018) and Nestin in TCGA datasets of lung adenocarcinoma and lung squamous cell carcinoma? Interrogation of larger datasets may provide additional evidence to support the conclusions drawn. Does Nestin expression have prognostic significance in human lung cancer (ADC and/or SCC)?

9. The bioactivity of 17-AAG is tightly regulated by NQO1, and thus cancer cells exhibiting perturbed Nrf2 pathway activation (due to alterations in KEAP1 and/or NRF2) have shown to exhibit increased sensitivities to 17-AAG treatment. Drug titration/sensitivity assays (MTA/Cell Titre GLO) in the individual cell lines (in vitro) would be of added value, and complement other findings presented. Should ovNestin + PBS also be included as an additional control, given that these tumours appear to grow larger than control tumours (Figure 7)?

Minor comments:

1. Results section entitled "The ESGE motif in Nestin is essential for its ability to bind the Kelch domain of Keap1" – spelling error, paragraph 3. Demonstrated – should be altered to "demonstrated"

Reviewer #2 (Remarks to the Author):

The study reports on a rather surprising finding that nestin would be critically involved as survival determinant in lung cancer. Nestin has previously been reported as a modulator of differentiation in muscle cells and neuronal cells, but there are no indications of involvement in lung cancer.

This study reports binding of a specific motif in nestin to the kelch-domain of Keap1, thereby stabilizing Nrf2, as it escapes binding to Keap1.

In a way, this study is a continuation of a recent study by this group, where they showed the involvement of nuclear nestin in tumor senescence by regulating Cdk5, which in turn regulated nuclear structure (Zhang et al. Nature Communications 2018).

This is a mechanistically elegant study, which supports the presented hypothesis with very strong data sets. It certainly also significantly advances our knowledge of nestin-mediated functions and adds to the cancer dimension of nestin.

The complete absence of Cdk5 in their presentation comes across as surprising. Given their previous study, which pointed out a clear interrelationship between nestin and Cdk5 activity, with clear consequences for cell proliferation and senescence, checking out a potential nestin-Cdk5 connection in the studied effects would have been a logical continuation. This would have been warranted also in light of a study already a long time ago (which is referred to; Sahlgren et al. EMBO J. 2006), which pointed out that nestin protects neuronal precursor cells from the same kind of oxidative stress that was used in this study. Finally and importantly, a recent study showed that the Cdk5 promotes translocation of Nrf2 to the nucleus and induce transcription of antioxidant genes (Jimenez-Blasco et al. Cell Death and Differentiation 2015). The Cdk5 nestin paradigm would fit so nicely to their concept that it would have been logical to test whether it could fit the presented paradigm of nestin as a regulator of the Keap-Nrf2 pathway. This could be done without extensive additions to the experimental part. Otherwise, the manuscript is carefully prepared, mostly with rather extensive data sets.

Specific comments:

Fig. 1: The experiments are primarily knockdown experiments. Have the authors tried whether overexpression of nestin augments the oxidative stress protection?

Fig. 2: Does nestin affect the nuclear localization of Nrf2, which has been reported as important for its activation of antioxidant genes?

Response to Reviewers

The reviewers raised a number of constructive criticisms and suggestions. To fully address them, we performed several additional experiments as well as implementing considerable changes to the manuscript. As a result, we believe the manuscript is much stronger. We wish to take this opportunity to thank the reviewers for their valuable input. Below, we summarize the reviewers' comments, and describe point-by-point how we have addressed them.

Reviewer #1

Point 1: The title “Nestin regulates cellular redox homeostasis and drug resistance in lung cancer through the Keap1-Nrf2 feedback loop” does not faithfully reflect the aim findings of the paper. Further evidence would be required in order to state that Nestin plays a role in “drug resistance” in this setting.

Response: Just as the reviewer point out, the aim of this study focus on the mechanism that Nestin maintain the redox balance by regulating the stability of Nrf2 in NSCLC. Because recent studies propose that targeting the redox balance will be a potential anticancer strategies (**Nat Rev Drug Discov. 2013;12:931-47**), we then ask whether the chemotherapeutic agent Tanespimycin (17-AAG) triggers oxidative stress and possesses stronger cytotoxicity under the condition of Nestin depletion.

Although our results demonstrate that Nestin knockdown rendered NSCLC cells more sensitive to 17-AAG in a subcutaneous tumor model in vivo (**Fig. 8e-8g**), it do not fully identify the role of nestin participating drug resistance. To avoid the overstatement and diluting the focus of this paper, we would like to revise the title as “Nestin regulates cellular redox homeostasis in lung cancer through the Keap1-Nrf2 feedback loop”. We agree that this is a good idea and have planned experiments along the suggested directions.

Point 2: Two lung adenocarcinoma cell lines were used for this study; A549 and H1299. A549 cells are KEAP1-mutant while H1299 cells are KEAP1-WT. Can the authors please comment on what effect this mutation may have on the subsequent findings of the paper? Particularly with respect to the ability of Nestin to “compete” with Nrf2 for Keap1 binding, as the Keap1-Nrf2 interaction in these cells is likely to

be abnormally perturbed.

Response: This is a good point. Several studies had demonstrated **high frequency (>20%) of mutation of Keap1** in primary lung cancers. Moreover, sequencing of *KEAP1* of 12 lung cancer cell lines revealed homozygous mutations in A549, H460, H838 and H1435, whereas H1395 and H1993 were heterozygous for mutant allele (**PLoS Med. 2006;3:e420**). In order to elucidate the interaction of WT and mutation Keap1 with Nestin, as well as Nrf2, we chose **A549 (Keap1 G333C) and H1299 (Keap1 WT)** as experimental cell lines. The results in Figure 4D-E showed that mutant Keap1 still have the ability to bind with Nestin or Nrf2 in A549 cell line, indicate that mutations (G333C) in first Kelch domain of Keap1 do not result in complete dissociation with DLG and ETGE motifs in Nestin and Nrf2.

To further explain the competitive Keap1-binding relationship between Nestin and Nrf2, we performed additional experiments. Co-immunoprecipitation assay using MG132-treated NSCLC cells identified that Keap1 bound more ubiquitinated-Nrf2 after Nestin knockdown. Moreover, the ubiquitylation levels of Nrf2 were more abundant in H1299 than that in A549 (**Attached Figure 1A**). Furthermore, we measured the mRNA levels of the transcriptional targets of Nrf2, and found that most antioxidant genes expression were higher in A549 than that in H1299 (**Attached Figure 1B**). As a result, A549 cells were exhibited stronger anti-apoptotic capacity than H1299, when exposed to H₂O₂ (**Fig. 1b and 1c**). In addition, we performed CellTiter-Glo assays to measure the sensitivity of the individual cell lines to 17-AAG. The result showed that the half maximal inhibitory concentration (IC₅₀) of 17-AAG was higher for A549 than H1299 (**Supplementary Fig. 8a and 8b**). These results indicate that Nrf2 pathway is more active in A549 cells.

Therefore, although mutant Keap1 protein (G333C) could influence its affinity with Nrf2 and protect Nrf2 from ubiquitin-proteasome degradation to some degree the conclusion about Nestin competitively binding to Keap1 and preventing Nrf2 degradation is objective, no matter whether Keap1 protein is mutant or not. We revised the corresponding data of CoIP in **Fig. 4g** and added these points to the Discussion section.

Attached Figure 1:

(A) The competitive Keap1-binding relationship between Nestin and Nrf2 were performed in A549 and H1299 cells by Co-immunoprecipitation assay.

(B) The mRNA levels of the transcriptional targets of Nrf2, antioxidant genes, were measured by q-PCR.

Point 3: Figure 1 – Decreased expression of a panel of anti-oxidant genes are observed in both A549 and H1299 cells following Nestin knockdown. Are the expression changes shown in panel D and E statistically significant? The authors hypothesize that modulation of the anti-oxidant response may have therapeutic potential. Did the author’s observe effects on tumour growth in experiments outlined in I-L? What cell line is represented in panel K? Images of single channel stains: NQO1, HO-1 and GCLM should also be included in panel L.

Response: These are good points.

For the first concern, the expression changes of antioxidant genes shown in panel d and e were statistically significant. As suggested, we added the asterisk in the corresponding subgroup.

For the second concern (**Fig. 1i-l**), we would like to first clarify that we actually measured the tumor growth rates and volumes in response to Dox-induced Nestin silencing and found that xenograft tumors following Dox treatment grew much slower than that of without Dox treatment. We have not presented in the original manuscript due to space limitations. As suggested, we added the corresponding data in the revised manuscript and **Supplementary Fig. 2b**.

For the third concern, we feel sorry for missing the cell line information in panel k.

A549 cell line was used in the inducible RNA interference xenograft tumor model. As suggested, the detail information was added in the revised manuscript and figure legend.

As suggested, we also presented the single channel stains of NQO1, HO-1 and GCLM in the revised **Fig. 11**.

Point 4: Figure 3 – The activity of Nrf2 is tightly controlled by its subcellular localization. In the cytoplasm Nrf2 is degraded via the proteasome, through its interaction with KEAP1 and the CUL3 ubiquitination ligase complex. shNestin 2 was used to evaluate the cytoplasmic/nuclear localization of Nrf2 in protein fractionation studies. The selection of shNestin 2 for these experiments is unclear, given that in Figure 2 Nrf2 protein expression was most markedly reduced (absent) upon Nestin KD with shNestin 2. Did the author's carry out fractionation experiments in A549 cells following KD of Nestin using shNestin 1?

Response: We would like to first clarify that we have already performed the experiments using both shNestin1 and shNestin2 and observed the similar results that Nestin knockdown caused the higher cytoplasmic/nuclear localization of Nrf2, and re-expression of Nestin rescued its nuclear translocation. We have not presented the shNestin1 data because of the higher knockdown efficiency of shNestin2. As suggested, we added the corresponding data of shNestin1 group in the revised **Supplementary Figure 4d**.

Point 5: Further clarification is required in relation to the IP/pull-down experiments shown in Figure 4. Are these experiments performed with endogenous KEAP1? Given that KEAP1 is mutated in A549 cells, how does this effect the results presented?

Response: Thanks for your kind suggestion. In fact, the CoIP assays in **Fig. 4d-e** represented the results of interaction between Nestin and endogenous Keap1. In addition, more ubiquitinated-Nrf2 were bound with Keap1 after Nestin knockdown (**Fig. 4g**). These results showed that the mechanisms that Nestin competitively bound to Keap1 and inhibited Nrf2 degradation were pervasive no matter Keap1 protein was

mutant or not. In view of the importance of Keap1 mutation in lung cancer, we added these discussions in revised manuscript.

Point 6: FL-Nestin should be included as an additional control in rescue experiments outlined in panels D-I.

Response: Accordingly, we performed full length of Nestin as an additional control in the rescue experiments and found that full length Myc-Nestin could also reverse the decreased antioxidant capacity and prevent the Nestin-knockdown cells from H₂O₂-induced cell death, which were similar with the rescue effect of Nestin fragment N1295-1621. The corresponding data were added to the Results section and **Fig. 5d-i and Supplementary Fig. 5a-c.**

Point 7: Figure 6 – Panel A: Can a complete sequence alignment be included as supplementary information? Can the authors please comment on why only hARE1 and hARE2 were selected for functional assays? Direct binding of Nrf2 to the Nestin promoter is demonstrated using EMSAs. Do ChIP assays confirm these findings? Have the author's interrogated publicly available Nrf2 ChIP-seq datasets to assess whether Nestin also comes up as a direct target? Chorley et al., Nucleic Acids Res 2012; Malhotra et al., Nucleic Acids Res 2010.

Response: As suggested, the complete sequence alignment was added in **Supplementary Table 1.** In fact, we have analyzed the location of these five hAREs. As hARE4 and hARE5 were resided in -8000bp of the promoter of Nestin, which seemed too far away from the transcription start sites of genes. Thus, hARE1, hARE2 and hARE3 were selected for further functional assays. We firstly found hARE1 or hARE2 of the Nestin promoter, rather than hARE3, enhanced the luciferase activity, after treating the NSCLC cells with the Nrf2 activator tBHQ (**Fig. 6b**). We then confirmed the direct binding of Nrf2 to the Nestin promoter at hARE1 and hARE2 sites using EMSA (**Fig. 6c and 6d**). We have not presented the data of hARE3 luciferase activity due to space limitations. To make this point more clear, we added this information to the Results section and the corresponding data was added as **Fig. 6.**

Accordingly, we interrogated publicly available datasets of Nrf2 ChIP-seq. However, Nestin was not listed as a direct target of Nrf2 (**Nucleic Acids Res. 2012;40:7416-29; Nucleic Acids Res. 2010;38:5718-34**). We speculate the possible reasons include: 1. Nestin had been identified as a member of intermediate filaments with tissue-specific expression patterns in adult stem cells or tumors (**Cold Spring Harb Perspect Biol. 2017;9.pii: a022046 ; Cell Mol Life Sci. 2018;75:2177-2195**). However, Chorley et al and Malhotra et al performed ChIP-seq assays using lymphoblastoid cell lines (LCLs) and mouse embryonic fibroblasts (MEFs), which were terminally differentiated cells with relatively low expression of Nestin. Therefore, the abundance of binding with Nestin promoter fragment might be lower than classical Nrf2 binding target, such as NQO1, HMOX1 in LCLs and MEFs.

2. It has been known that transcription factor binding to gene promoter required some specific conditions (**J Biol Chem. 2008;283:33554-33562; Nat Rev Mol Cell Biol. 2018;19:4-19**). NSCLC, as a malignant tumor, possessed some abnormally constitutively activated signaling pathway, such as EGFR, K-ras, which may lead to the specific conditions for the Nrf2-induced Nestin expression. In addition, we performed ChIP assay and found that direct binding of Nrf2 to the Nestin promoter using NSCLC cell lines (A549 and H1299), indicating that Nestin may be a novel transcriptional target of Nrf2. We added this information to the Results section and the corresponding data was presented as **Supplementary Fig. 6**.

Point 8: Correlations between Nrf2 and Nestin expression have been assessed in a panel of 200 NSCLC specimens. Have the authors correlated the expression of NQO1 (a robust biomarker of Nrf2 activity; Romero et al., Nature Medicine 2017; Best et al., Cell Metabolism 2018) and Nestin in TCGA datasets of lung adenocarcinoma and lung squamous cell carcinoma? Interrogation of larger datasets may provide additional evidence to support the conclusions drawn. Does Nestin expression have prognostic significance in human lung cancer (ADC and/or SCC)?

Response: As suggested, we performed additional analysis to address the reviewer's concerns. Using the website which is based on the TCGA databases (<http://gepia.cancer-pku.cn/index.html>, **Nucleic Acids Res. 2017;45:W98-W102**).

However, the expression of NQO1 was not significantly correlated with Nestin (**Attached Figure 2A**).

For the prognostic significance of Nestin expression, we analyzed the Overall Survival (OS) and Disease Free Survival (DFS) in the two groups of 15% cutoff high and 15% cutoff low Nestin expression. Interestingly, Kaplan-Meier OS and DFS curves of the TCGA lung cancer cohort show that patients with high Nestin level had a worse prognosis than patients with low Nestin level (OS: log-rank $P=0.042$, HR [high] =1.5; DFS: log-rank $P=0.021$, HR [high] =1.7). In detail, when Nestin was high expression, the patients with LUAD (Lung adenocarcinoma) or LUSC (Lung squamous cell carcinoma) represented significantly poor OS (log-rank $P=0.012$, HR [high] =2.1) or DFS rates (log-rank $P=0.042$, HR [high] =2), respectively (**Attached Figure 2B**). In conclusion, these results fully demonstrated that Nestin expression was closely correlated with poor survival and could be used as a novel prognostic biomarker for patients with lung cancer. The corresponding data were added as **Supplementary Fig. 7**.

Attached Figure 2. The clinical prognostic significance of Nestin expression based on TCGA databases.

(A) The correlation between the expression of NQO1 with Nestin in TCGA datasets of lung adenocarcinoma (LUAD) and lung squamous cell carcinoma (LUSC).

(B) Overall Survival (OS) and Disease Free Survival (DFS) in the two lung cancer patients groups of 15% cutoff high and 15% cutoff low Nestin expression, were analyzed using the GEPIA website (<http://gepia.cancer-pku.cn/index.html>).

Point 9: The bioactivity of 17-AAG is tightly regulated by NQO1, and thus cancer cells exhibiting perturbed Nrf2 pathway activation (due to alterations in KEAP1 and/or NRF2) have shown to exhibit increased sensitivities to 17-AAG treatment. Drug titration/sensitivity assays (MTA/Cell Titre GLO) in the individual cell lines (in

vitro) would be of added value, and complement other findings presented. Should ovNestin + PBS also be included as an additional control, given that these tumours appear to grow larger than control tumours (Figure 7)?

Response:

As suggested, we performed CellTiter-Glo assays to measure the sensitivity of the individual cell lines to 17-AAG. A549 or H1299 cell lines were treated with increasing concentrations of 17-AAG for 48 h, respectively. The results showed that 17-AAG significantly induced the apoptosis of NSCLC cell lines in a dose-dependent manner. The IC50 of 17-AAG was about 85.2 nM for A549 cells and 71.8 nM for H1299 cells. Moreover, Nestin knockdown increased the sensitivity of NSCLC cells to 17-AAG, while Nestin overexpression increased the viability of 17-AAG treated tumor cells. The corresponding data were added as **Supplementary Fig. 8a-d**.

For the reviewer's second concern, given we have already demonstrated overexpression of Nestin could promote tumor growth in the xenograft model (ovNestin group, **Fig. 7j-k**) and further confirmed that Nestin overexpression antagonized the cytotoxicity of 17-AAG on NSCLC cells through the bioluminescent imaging (**Fig. 8e-g**), we feel ovNestin + PBS xenograft control group might not be the priority of the current scope of this paper in consideration of the amount of time that these follow-up studies will likely require and the animal welfare.

Minor point 1: Results section entitled "The ESGE motif in Nestin is essential for its ability to bind the Kelch domain of Keap1" – spelling error, paragraph 3. Demonstrated – should be altered to "demonstrated".

Response: We apologize for any errors that were due to our oversight during the preparation of the original manuscript. We hope that the care we have exercised in the revision would have resulted in a more improved manuscript with far fewer mistakes.

Reviewer #2

Point 1: The complete absence of Cdk5 in their presentation comes across as surprising. Given their previous study, which pointed out a clear interrelationship between nestin and Cdk5 activity, with clear consequences for cell proliferation and senescence, checking out a potential nestin-Cdk5 connection in the studied effects would have been a logical continuation. This would have been warranted also in light of a study already a long time ago (which is referred to; Sahlgren et al. *EMBO J.* 2006), which pointed out that nestin protects neuronal precursor cells from the same kind of oxidative stress that as used in this study. Finally and importantly, a recent study showed that the Cdk5 promotes translocation of Nrf2 to the nucleus and induce transcription of antioxidant genes (Jimenez-Blasco et al. *Cell Death and Differentiation* 2015). The Cdk5 nestin paradigm would fit so nicely to their concept that it would have been logical to test whether it could fit the presented paradigm of nestin as a regulator of the Keap-Nrf2 pathway. This could be done without extensive additions to the experimental part. Otherwise, the manuscript is carefully prepared, mostly with rather extensive data sets.

Response: This is a good point, which could deepen our understandings of the regulatory mechanisms between Nestin and Nrf2. Despite the aforementioned evidence for the regulation of Nrf2 via Keap1, there are several clues showing that Nrf2 can be phosphorylated and translocate into nucleus independently of Keap1, such as by PKC, CK2, Cdk5 and so on (***Pharmacol Rev.* 2018;70:348-383; *Biochem Pharmacol.* 2013;85:705-17; *Cell Death Differ.* 2015;22:1877–1889**). Therefore, we hypothesized that Nrf2 localized and degraded in the cytoplasm was due to reduced Cdk5-mediated Nrf2 phosphorylation after Nestin knockdown. However, Sahlgren et al. demonstrated downregulation of Nestin subsequently induced the activation of Cdk5 in neuronal precursor cells under oxidative stress (***EMBO J.* 2006;25:4808–4819**). Similarly, in our previous studies, we found that the protein levels of Cdk5 were unchanged, but its activity was significantly increased in Nestin-knockdown NSCLC cells (**Fig. 6f, *Nat Commun.* 2018;9:3613**). Thus, it seemed contradictory that depletion of Nestin leading to Nrf2 degradation was caused by reduced Cdk5 phosphorylation activity. Accordingly, we performed CoIP assays found that the

phosphorylation levels of Nrf2 were decreased after Nestin knockdown (**Attached Figure 3**). In conclusion, it seems that Cdk5 might not be involved in the signaling pathway of Nestin on protecting Nrf2 from Keap1-mediated degradation.

Attached Figure 3. The phosphorylation levels of Nrf2 were decreased after Nestin knockdown in NSCLC cells using CoIP assay.

Minor Point 1: Fig. 1: The experiments are primarily knockdown experiments. Have the authors tried whether overexpression of nestin augments the oxidative stress protection?

Response: As suggested, we measured the mRNA expression changes of antioxidant genes (CAT, GPX1, GPX4, SOD1, SOD2, GCLC, GCLM, HMOX1 and NQO1) in two NSCLC cells lines transfected with Myc-Nestin. The results showed that Nestin overexpression could enhance the expression of most these genes in both A549 and H1299 cells at transcriptional levels (**Supplementary Fig. 1g-h**). Moreover, the levels of the GSH, the activities of SOD and CAT, and the total antioxidant capacity were all increased upon Nestin overexpression (**Supplementary Fig. 1i-k**). We added the corresponding data in the revised supplementary files.

Minor Point 1: Fig. 2: Does nestin affect the nuclear localization of Nrf2, which has been reported as important for its activation of antioxidant genes?

Response: Thanks for your kind suggestion. Actually, we have presented the results of Nestin expression on the nuclear localization of Nrf2 in **Fig. 3g**. Using Western

Blot assay, we showed that Nestin-knockdown caused more cytoplasmic distribution of Nrf2, and the rescue experiment demonstrated the re-expression of Nestin restored the nuclear translocation of Nrf2.

REVIEWERS' COMMENTS:

Reviewer #1 (Remarks to the Author):

I thank the authors for careful consideration of my concerns. They have performed additional experiments and analysis to address these comments, which strengthen the conclusions of the manuscript. In my opinion, the manuscript is suitable for publication in Nature Communications.

Major comments:

The authors extend their analysis to include data from the TCGA. However, in contrast to findings from 200 NSCLC patients (Figure 8) no significant correlation is observed between Nestin mRNA expression and NQO1 (rebuttal Figure 2A). How do the authors reconcile these differences? Did they extend their analysis to include HO-1, GCLM and GCLC? Does this impact on the conclusions drawn from this part of the study? This discrepancy is worth discussing in the manuscript.

Minor comments:

1. Line 115: "need" should be altered to "needs"
2. Line 118-119: "Keap1-mediated degradation subsequently upregulate...", should be altered to "Keap1-mediated degradation and subsequently upregulates..."
3. Line 449: KEAP1 should be italics
4. Line 450: KEAP1 should be italics
5. Line 452: Keap1 should be italics

Reviewer #2 (Remarks to the Author):

The authors have thoroughly addressed the issues raised by the individual reviewers. I am also satisfied with the response and reasoning related to the Cdk5 question. However, since this is such an obvious issue, as it was raised in their previous study, which is also used a reference point for this study, I would urge the authors to add a few sentences to the discussion and mention that this possibility has been considered and addressed. They can even refer to data not shown. Adding such a short statement would also more tightly link this study as a logical continuum to their previous study. Easily one forgets that negative data can also be important data to present.

Response to Reviewers

We thank all the reviewers for their time and positive feedback. Below, we summarize the reviewers' comments, and describe point-by-point how we have addressed them.

Reviewer #1 (Remarks to the Author):

I thank the authors for careful consideration of my concerns. They have performed additional experiments and analysis to address these comments, which strengthen the conclusions of the manuscript. In my opinion, the manuscript is suitable for publication in Nature Communications.

Major comments:

The authors extend their analysis to include data from the TCGA. However, in contrast to findings from 200 NSCLC patients (Figure 8) no significant correlation is observed between Nestin mRNA expression and NQO1 (rebuttal Figure 2A). How do the authors reconcile these differences? Did they extend their analysis to include HO-1, GCLM and GCLC? Does this impact on the conclusions drawn from this part of the study? This this discrepancy is worth discussing in the manuscript.

Response: Thank you for your kind suggestion. Although the expression of NQO1 was not significantly correlated with Nestin, we extend our analysis about other Nrf2 target genes and found that the expression of HMOX1 and GCLM were significantly correlated with Nestin based on TCGA database (**Attached Figure 1A**), which was in accordance with our findings. In our opinion, the differences of individual patients or the specimen collection and processing might be partly account for the difference between our clinical data and TCGA database. In conclusion, this did not impact on the conclusions drawn from our study.

Attached Figure 1.

(A) The correlation between the expression of HMOX1 and GCLM with Nestin in TCGA datasets of lung adenocarcinoma (LUAD) and lung squamous cell carcinoma (LUSC), respectively.

Minor comments:

1. Line 115: “need” should be altered to “needs”
2. Line 118-119: “Keap1-mediated degradation subsequently upregulate...”, should be altered to “Keap1-mediated degradation and subsequently upregulates...”
3. Line 449: KEAP1 should be italics
4. Line 450: KEAP1 should be italics
5. Line 452: Keap1 should be italics

Response: We feel sorry for our oversight during preparation of manuscript. As suggested, these mistakes had been revised in the recent manuscript.

Reviewer #2 (Remarks to the Author):

The authors have thoroughly addressed the issues raised by the individual reviewers. I am also satisfied with the response and reasoning related to the Cdk5 question. However, since this is such an obvious issue, as it was raised in their previous study, which is also used a reference point for this study, I would urge the authors to add a few sentences to the discussion and mention that this possibility has been considered and addressed. They can even refer to data not shown. Adding such a short statement would also more tightly link this study as a logical continuum to their previous study. Easily one forgets that negative data can also be important data to present.

Response: Thank you for your kind suggestion. We had added the discussion about the relationship between Nestin and Cdk5 in the regulation of Nrf2 in the Discussion Section.